# CPT1A mediates radiation sensitivity in colorectal cancer

Zhenhui Chen[1†], Lu Yu[2†], Zhihao Zheng[2†], Xusheng Wang[2], Qiqing Guo[2], Yuchuan Chen[3], Yaowei Zhang[2], Yuqin Zhang[2], Jianbiao Xiao[4], Keli Chen[5*], Hongying Fan[1*], Yi Ding[2*]

[1]Department of Microbiology, Guangdong Provincial Key Laboratory of Tropical Disease Research, School of Public Health, Southern Medical University, Guangzhou, China; [2]Department of Radiation Oncology, Nanfang Hospital, Southern Medical University, Guangzhou, China; [3]State Key Laboratory of Organ Failure Research, Key Laboratory of Infectious Diseases Research in South China, Ministry of Education, Guangdong Provincial Key Laboratory of Viral Hepatitis Research, Guangdong Provincial Clinical Research Center for Viral Hepatitis, Department of Infectious Diseases, Nanfang Hospital, Southern Medical University, Guangzhou, China; [4]Department of Pathology, Nanfang Hospital and School of Basic Medical Science, Southern Medical University, Guangzhou, China; [5]HuiQiao Medical Center, Nanfang Hospital, Southern Medical University, Guangzhou, China

*For correspondence:
16474801@qq.com (KC);
gzfhy@smu.edu.cn (HF);
dy512@smu.edu.cn (YD)

†These authors contributed equally to this work

**Competing interest:** The authors declare that no competing interests exist.

## eLife Assessment

This study reports a **valuable** finding for the treatment of colorectal cancer (CRC), as the authors demonstrated that the enzyme CPT1A plays an significant role in the response to radiotherapy in CRC patients. However, the reviewers found that the results presented are still **incomplete**.

**Abstract** The prevalence and mortality rates of colorectal cancer (CRC) are increasing worldwide. Radiation resistance hinders radiotherapy, a standard treatment for advanced CRC, leading to local recurrence and metastasis. Elucidating the molecular mechanisms underlying radioresistance in CRC is critical to enhance therapeutic efficacy and patient outcomes. Bioinformatic analysis and tumour tissue examination were conducted to investigate the *CPT1A* mRNA and protein levels in CRC and their correlation with radiotherapy efficacy. Furthermore, lentiviral overexpression and CRISPR/Cas9 lentiviral vectors, along with in vitro and in vivo radiation experiments, were used to explore the effect of CPT1A on radiosensitivity. Additionally, transcriptomic sequencing, molecular biology experiments, and bioinformatic analyses were employed to elucidate the molecular mechanisms by which CPT1A regulates radiosensitivity. CPT1A was significantly downregulated in CRC and negatively correlated with responsiveness to neoadjuvant radiotherapy. Functional studies suggested that CPT1A mediates radiosensitivity, influencing reactive oxygen species (ROS) scavenging and DNA damage response. Transcriptomic and molecular analyses highlighted the involvement of the peroxisomal pathway. Mechanistic exploration revealed that CPT1A downregulates the FOXM1-SOD1/SOD2/CAT axis, moderating cellular ROS levels after irradiation and enhancing radiosensitivity. CPT1A downregulation contributes to radioresistance in CRC by augmenting the FOXM1-mediated antioxidant response. Thus, CPT1A is a potential biomarker of radiosensitivity and a novel target for overcoming radioresistance, offering a future direction to enhance CRC radiotherapy.

## Introduction

Colorectal cancer (CRC) is the second-highest cause of cancer-related mortality (*Siegel et al., 2023*). Radiotherapy is crucial for CRC management, especially in patients with locally advanced rectal cancer (cT$_{3-4}$N$_+$) (*Glynne-Jones et al., 2017*). Neoadjuvant therapies show clinical or pathological complete response in 16–30% of patients, realising downstaging in approximately 60% of patients, significantly enhancing local control, and facilitating curative surgery (*Cercek et al., 2018*). Moreover, radiotherapy is beneficial in initially unresectable and recurrent cases with limited metastasis to organs, such as the liver and lungs (*Cervantes et al., 2023*). Nonetheless, the effectiveness of radiotherapy is affected by radioresistance, which precipitates tumour relapse and metastasis and currently lacks an efficacious clinical resolution. Unlocking the molecular mechanisms underlying CRC radioresistance will enhance outcomes and improve patient prognoses.

Reactive oxygen species (ROS) are byproducts of normal cellular metabolism occurring in organelles, such as mitochondria, endoplasmic reticulum, and peroxisomes (*Bhattacharyya et al., 2014*). Within mitochondria, 90% of cellular ROS are generated by complexes I, II, and IV of the electron transport chain (*Glasauer and Chandel, 2014*). Exogenous stimuli, including radiation, can cause significant, acute elevations in ROS levels (*Hecht et al., 2024*). ROS function ambivalently in the intracellular signalling and redox homoeostasis of tumour cells (*Shah et al., 2024*). ROS amplify oncogenic phenotypes, such as proliferation and differentiation, hasten the accumulation of metastasis-inducing mutations, and aid tumour cell survival under hypoxic conditions (*Palma et al., 2024*). However, excess ROS precipitate apoptosis and other cell death types from oxidative stress (*Palma et al., 2024*). Thus, intracellular ROS generation is meticulously monitored and regulated by a comprehensive ROS-scavenging system encompassing antioxidants and antioxidative enzymes (*Palma et al., 2024*; *Shah et al., 2024*).

Radiation ionises water molecules, creating an intracellular surge of ROS, which indirectly cause two-thirds of DNA damage (*Chio and Tuveson, 2017*). Consequently, ROS scavenging inevitably influences cancer cell radiosensitivity (*Skvortsova et al., 2015*). Increased expression and activity of antioxidative enzymes, such as peroxidase (POD), catalase (CAT), glutathione peroxidase, and glutathione reductase, are correlated with radiosensitivity (*Hecht et al., 2024*).

Carnitine palmitoyltransferase 1 (CPT1) is an outer mitochondrial membrane that catalyses the rate-limiting step of fatty acid oxidation and is absent in several tumours (*Melone et al., 2018*). The CPT1 family contains three isoforms: CPT1A, CPT1B, and CPT1C. Research on CPT1A has been detailed (*Schlaepfer and Joshi, 2020*). CPT1A is critical to cancer cell growth, survival, and drug resistance, making it an attractive target (*Qu et al., 2016*). CPT1A also interacts with other key pathways and factors regulating gene expression and apoptosis in cancer cell (*Qu et al., 2016*). However, its role in CRC and radiotherapy resistance is unclear.

Previously, we found that various metabolic pathways, including fatty acid metabolism, are closely related to tumour radioresistance. Therefore, this study mainly focused on CPT1A, which affects CRC radiosensitivity, to reveal novel therapeutic strategies to mitigate radiotherapy resistance and improve clinical outcomes.

## Results

### Low expression of CPT1A in CRC tumours

Analysis of CRC mRNA sequencing arrays from GEO consistently indicated low *CPT1A* mRNA levels in CRC (*Figure 1A*). The transcript levels of *CPT1A* were lower in most colon cancers (19/24 pairs) than in the adjacent non-tumour tissues (*Figure 1B*). Meanwhile, CPT1A protein levels were lower in most CRC tissues (14/16 pairs) than in the adjacent non-tumour tissues (*Figure 1C*). Further exploration of CPT1A expression in TCGA revealed that CPT1A mRNA levels were significantly lower in colon adenocarcinoma (COAD) tissues than in the adjacent non-tumour tissues (*Figure 1D*). Similar results were observed for rectal adenocarcinoma (READ) (*Figure 1E*). Immunohistochemistry (IHC) staining of the cancer-adjacent borders of two patients showed that CPT1A protein levels were lower in CRC tissues than in the nearby non-tumour tissues (*Figure 1F*). These findings provide comprehensive evidence supporting the downregulation of *CPT1A* expression at both mRNA and protein levels in CRC.

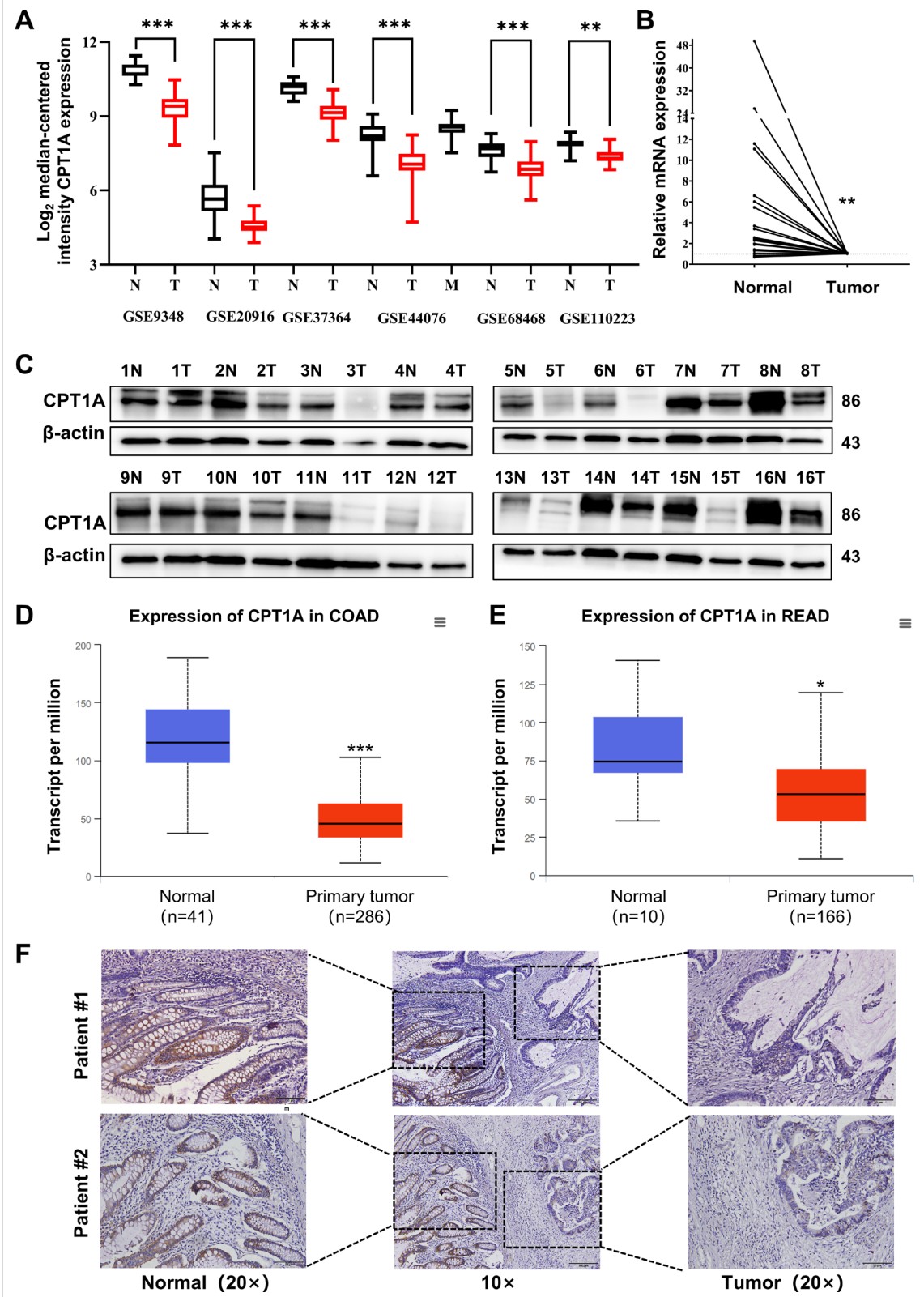

**Figure 1.** Aberrant CPT1A mRNA level in colorectal cancer (CRC). (**A**) The expression of CPT1A in six GEO microarrays. (**B**) Real-time PCR for CPT1A in 24-paired CRC and adjacent non-tumour tissues. (**C**) Western blot for CPT1A in 16-paired CRC and adjacent non-tumour tissues. (**D**) Lower CPT1A mRNA level in colon adenocarcinoma (COAD) than the normal counterparts from TCGA in UALCAN database. (**E**) Lower CPT1A mRNA level in rectal

*Figure 1 continued on next page*

*Figure 1 continued*

adenocarcinoma (READ) than the normal counterparts from TCGA in UALCAN database. (**F**) Immunohistochemistry (IHC) assay for CPT1A in two patients, scale bar = 100µm. ***p<0.001, **p<0.01, *p<0.05.

The online version of this article includes the following source data for figure 1:

**Source data 1.** Original files for western blot analysis displayed in *Figure 1C*.

**Source data 2.** PDF file containing original western blots for *Figure 1C*.

## Low CPT1A CRC exhibits radioresistance and poor overall survival

Based on the IHC scores, patients were divided into CPT1A high- and low-expression groups (*Figure 2A*). Kaplan-Meier survival analysis indicated that low CPT1A-expressing patients had low overall survival (OS) (*Figure 2B*). Survival analysis of READ patients with low CPT1A using GEPIA also suggested a low OS; the difference was statistically close but not significant (p=0.061, *Figure 2C*). To

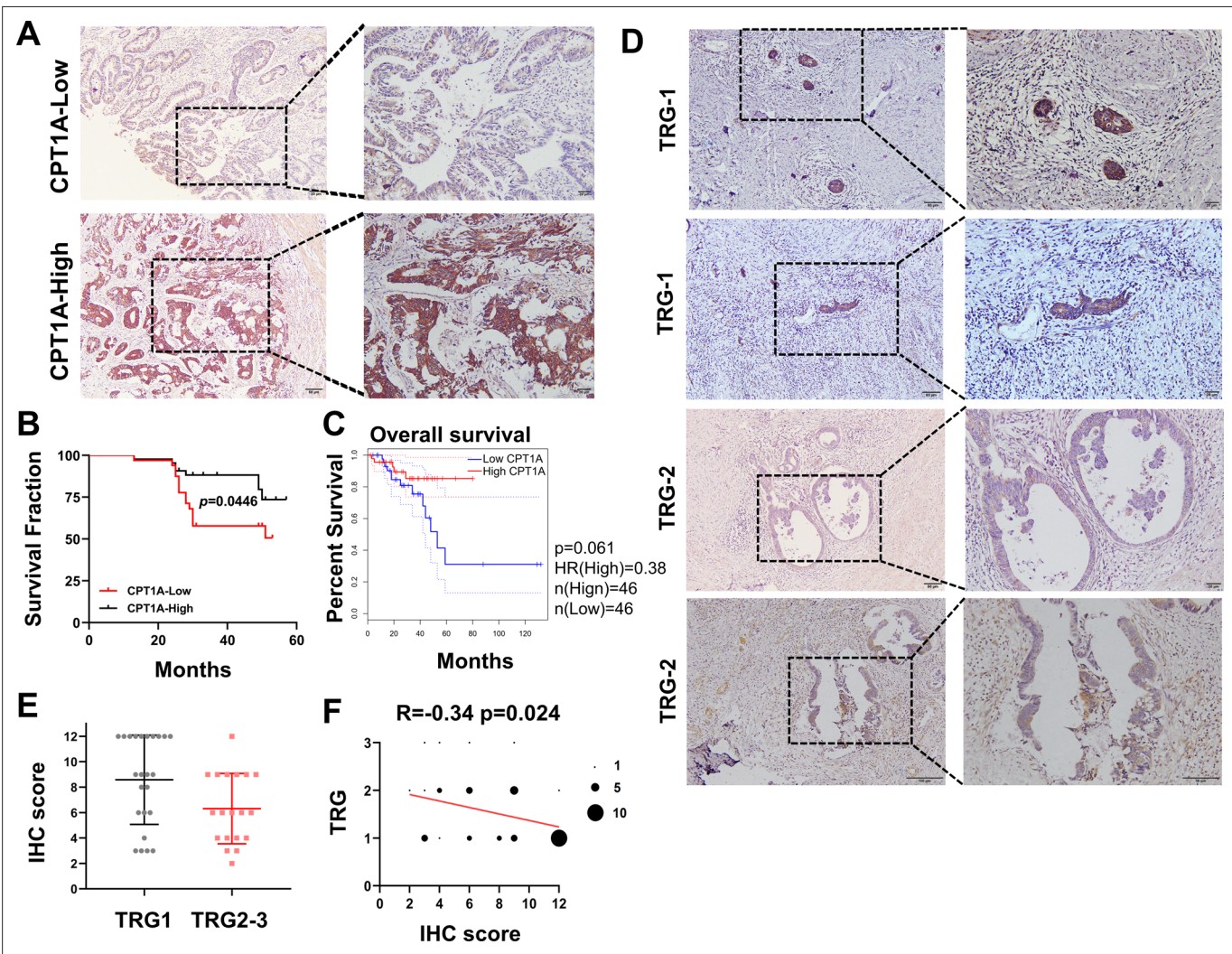

**Figure 2.** Correlation of CPT1A with overall survival (OS) and neoadjuvant therapy response in rectal cancer patients. (**A**) Immunohistochemistry (IHC) assay for CPT1A in two groups of patients, upper with low CPT1A expression and lower with high CPT1A expression (IHC score≥6). (**B**) The OS was estimated by the Kaplan-Meier method in rectal cancer patients with low (n=33) or high CPT1A expression (n=43). (**C**) The OS was estimated by the Kaplan-Meier method in rectal cancer patients in TCGA database with low (n=46) or high CPT1A expression (n=46). (**D**) IHC assay for CPT1A in two groups of patients, upper with TRG-1 and lower with TRG-2 (TRG means tumour regression grade, AJCC standard, 0, complete response: no remaining viable cancer cells; 1, moderate response: only small clusters or single cancer cells remaining; 2, minimal response: residual cancer remaining, but with predominant fibrosis), scale bar = 100µm. . (**E**) Dot plot showing the IHC score and TRG score of patients. (F) Correlation of CPT1A with TRG score, size of dot represents the number. ***p<0.001, **p<0.01, *p<0.05.

further investigate the correlation between CPT1A and radiotherapy efficacy, IHC staining and scoring for CPT1A were performed on samples from 43 patients with rectal cancer who received neoadjuvant radiochemotherapy. Among patients with a tumour regression grade (TRG) score of 1, which indicates a minimal number of residual tumour cells, a high IHC score for CPT1A is often observed. Conversely, patients with a TRG score of 2, indicating more residual tumour cells, exhibited relatively lower IHC staining intensity and overall scores (*Figure 2D*). A comparison of groups with TRG 2–3 and TRG 1 revealed significantly higher IHC scores in the TRG 1 group (*Figure 2E*). The TRG and IHC scores for CPT1A showed a negative correlation (R=−0.3430 and p=0.024) (*Figure 2F*). In summary, low CPT1A expression was associated with poor OS, high TRG scores, and a high probability of radioresistance.

## Decreased CPT1A expression contributes to radiation resistance in CRC cells

Through a colony formation assay (CFA) and multi-target single-hit survival model, we found that SW480, Caco-2, SW620, HT-29, and cells were more resistant to radiation than HCT15, RKO, and HCT 116 cells (*Figure 3—figure supplement 1A and B*, *Supplementary file 1*). Furthermore, the background expression levels of CPT1A in the above cells revealed that CPT1A transcription and protein levels were higher in radiation-resistant cells than in radiosensitive cells (*Figure 3—figure supplement 1C and D*).

Accordingly, we constructed stable CRC cell lines with *CPT1A* knockout/overexpression. We transfected the CRISPR/Cas9 lentivirus into HCT 116 cells (highest CPT1A expression and radiosensitive) and used western blotting to verify that the knockout efficiency of the second site was the highest, while mRNA levels were significantly reduced (*Figure 3A and B*). We used the second knockout site for subsequent in vitro and in vivo experiments. We also transfected the CPT1A-overexpressing lentivirus into SW480 cells (lowest CPT1A expression and radioresistant) and verified that CPT1A protein and mRNA levels increased (*Figure 3A and B*).

The CFA and multi-target single-hit survival model suggested that radioresistance increased with *CPT1A* knockout (D0=1.526 vs 1.993, p<0.05, *Figure 3C and D*), whereas the radioresistance of cells decreased with *CPT1A* overexpression (D0=2.724 vs 1.963, p<0.01, *Figure 3E and F*). The comet assay suggested that the proportion of DNA in the tail of cells decreased with CPT1A knockout, indicating improved damage repair. With *CPT1A* overexpression, the proportion of DNA in the tail increased, indicating reduced damage repair capabilities (*Figure 3G*). We also detected γ-H2A.X expression at different time points after 6 Gy irradiation; with *CPT1A* knockout, γ-H2A.X disappeared from cells faster, indicating improved cell damage repair (*Figure 3H*). With *CPT1A* overexpression, the disappearance rate of γ-H2A.X in cells was slower, remaining detectable even after 24 hr, indicating diminished cellular repair capability (*Figure 3I*). Collectively, our data suggest that CPT1A radiosensitises intrinsically radioresistant cells.

We generated radioresistant cell lines (HCT-15-25F and HCT-15-5F) from the HCT-15 parent line by fractionated irradiation (*Figure 3—figure supplement 2A*). Using CFAs and the multi-target single-hit survival model, we observed increased radioresistance in these new cell lines, as indicated by higher D0 values than those of the parental cells (HCT-15 D0=2.957, HCT-15-5F D0=3.240, HCT-15-25F D0=3.822) (*Figure 3—figure supplement 2B and C*). We also found significantly decreased CPT1A protein expression in HCT-15-25F and HCT-15-5F cells compared with that in the original cells (*Figure 3—figure supplement 2D*). Thus, HCT-15-25F cells were selected for further analysis. Transfection with the *CPT1A*-overexpressing lentivirus led to stable overexpression in HCT-15-25F cells (*Figure 3—figure supplement 2E*). To assess the impact of *CPT1A* overexpression on DNA repair capacity, we monitored γ-H2A.X expression at different time points following 6 Gy irradiation. *CPT1A* overexpression resulted in a slow disappearance of intracellular γ-H2A.X, suggesting enhanced DNA damage repair capability (*Figure 3—figure supplement 2F*). CFAs and the multi-target single-hit survival model revealed that *CPT1A* overexpression increased the radiosensitivity of radioresistant cells (D0=2.871 vs 2.581, p<0.05; *Figure 3—figure supplement 2G and H*), indicating that CPT1A exerts a radiosensitising effect in inducible radioresistant cell lines.

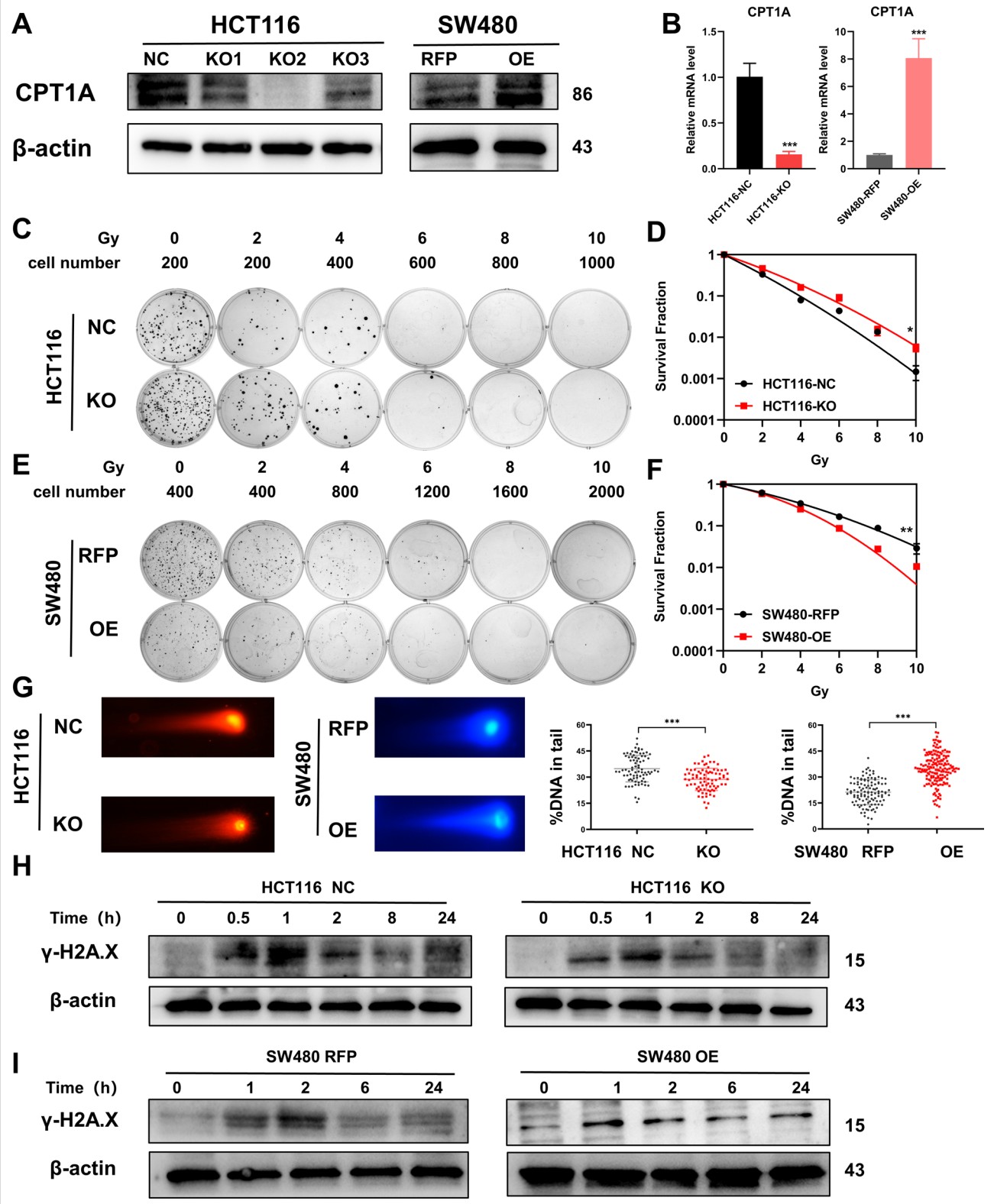

**Figure 3.** Radiosensitivity of stable knockout or overexpression of CPT1A. (**A**) The protein level of CPT1A in HCT 116-NC, HCT 116-KO1, HCT 116-KO2, HCT 116-KO3, SW480-RFP, and SW480-OE cell lines. (**B**) The mRNA level of CPT1A in HCT 116-NC, HCT 116-KO (KO2), SW480-RFP, and SW480-OE cell lines. (**C**) Colony-forming assay of HCT 116-NC and HCT 116-KO cell lines. (**D**) The map of multi-target single-hit model. (**E**) Colony-forming assay of SW480-RFP and SW480-OE cell lines. (**F**) The map of multi-target single-hit model. (**G**) Comet assay of HCT 116-NC, HCT 116-KO, SW480-RFP, and

*Figure 3 continued on next page*

*Figure 3 continued*

SW480-OE. (**H**) Protein expression of γ-H2A.X in HCT 116-NC and HCT 116-KO cell lines. (**I**) Protein expression of γ-H2A.X in SW480-RFP and SW480-OE cell lines. ***p<0.001, **p<0.01, *p<0.05.

The online version of this article includes the following source data and figure supplement(s) for figure 3:

**Source data 1.** Original files for western blot analysis displayed in *Figure 3*.

**Source data 2.** PDF file containing original western blots for *Figure 3*.

**Figure supplement 1.** The level of CPT1A in different colorectal cancer (CRC) cell lines.

**Figure supplement 1—source data 1.** Original files for western blot analysis displayed in *Figure 3—figure supplement 1*.

**Figure supplement 1—source data 2.** PDF file containing original western blots for *Figure 3—figure supplement 1*.

**Figure supplement 2.** Establishment of radioresistance cells and overexpression of CPT1A rescues the radiation resistance of HCT-15-25F cells.

**Figure supplement 2—source data 1.** Original files for western blot analysis displayed in *Figure 3—figure supplement 2*.

**Figure supplement 2—source data 2.** PDF file containing original western blots for *Figure 3—figure supplement 2*.

## Diminished CPT1A expression in vivo also leads to tumour radioresistance

To further validate the impact of CPT1A on radiation resistance in vivo, we established a xenograft model using HCT116-NC and HCT116-KO cell lines in nude mice (*Figure 4A*). *CPT1A*-stabilising knockout increased tumour weights in mice, which persisted after radiotherapy (*Figure 4B*), suggesting that *CPT1A* knockout promotes tumour growth and confers increased resistance to radiation. IHC staining demonstrated a significant increase in Ki-67 staining intensity and the percentage of positive cells in *CPT1A* knockout tumours, indicating enhanced proliferative capacity that was further pronounced after radiotherapy (*Figure 4C and D*). In the absence of CPT1A knockdown, radiotherapy reduced the percentage of Ki67-positive cells in the xenograft tumours by 32.9% (approximately 39.6% of the pre-irradiation baseline). In contrast, upon CPT1A knockdown, radiotherapy only led to a 14.5% reduction in the percentage of Ki67-positive cells (approximately 15.6% of the pre-irradiation baseline; *Figure 4D*). Similarly, we performed xenograft model experiments using the SW480-RFP and SW480-OE cell lines (*Figure 4E*). *CPT1A* overexpression resulted in reduced tumour weight in mice, a trend that persisted even after radiotherapy (*Figure 4F*). Furthermore, as illustrated in *Figure 4E and F*, in the absence of CPT1A overexpression, radiotherapy resulted in a 0.10 g decrease in tumour weight (approximately 52.5% of the pre-irradiation weight), whereas with CPT1A overexpression, radiotherapy induced a more pronounced 0.12 g reduction in tumour weight (approximately 89.7% of the pre-irradiation weight). These findings suggested that *CPT1A* overexpression inhibits tumour growth and sensitises tumour cells to radiation. IHC staining for Ki-67 revealed significantly decreased staining intensity and percentage of positive cells in CPT1A-overexpressing tumours, indicating weakened proliferative capacity, which was further accentuated after radiotherapy (*Figure 4G and H*).

## Low CPT1A levels accelerate post-radiation ROS scavenging

To study the mechanism of low CPT1A expression in radiotherapy resistance, we conducted differential gene expression analysis between HCT116-KO and HCT116-NC cells. The gene expression heatmap demonstrated high consistency among technical replicates for both HCT 116-NC and HCT 116-KO cells (*Figure 5—figure supplement 1A*). With *CPT1A* knockdown, we found 363 upregulated and 1290 downregulated genes ($|\log_2(\text{fold change})|>1$ and q<0.05) (*Figure 5—figure supplement 1B*). We conducted Kyoto Encyclopedia of Genes and Genomes (KEGG) pathway analysis and Gene Ontology (GO) annotation for all differentially expressed genes (DEGs; *Figure 5A*; *Figure 5—figure supplement 1C–E*), showing that the peroxisomes, cell cycle nucleotide excision repair, and fatty acid degradation pathways were among the enriched KEGG pathways (*Figure 5A*). Peroxisomes are important organelles that maintain cellular redox balance by clearing ROS. Therefore, we targeted peroxisomal pathways. ROS levels in the cells were dynamically balanced, including ROS production and scavenging (*Figure 5B*). To investigate the effect of CPT1A on ROS, we examined them in stable *CPT1A* knockout/overexpression cells following 6 Gy irradiation and 1 hr of incubation with 2',7'-dichlorodihydrofluorescein diacetate (DCFH-DA), a fluorescent redox probe used to detect ROS in cells (*Figure 5C*). The total ROS levels in *CPT1A* knockout cells decreased, whereas those in *CPT1A*-overexpressing cells increased (*Figure 5D*). The main mechanism by which cells produce ROS under

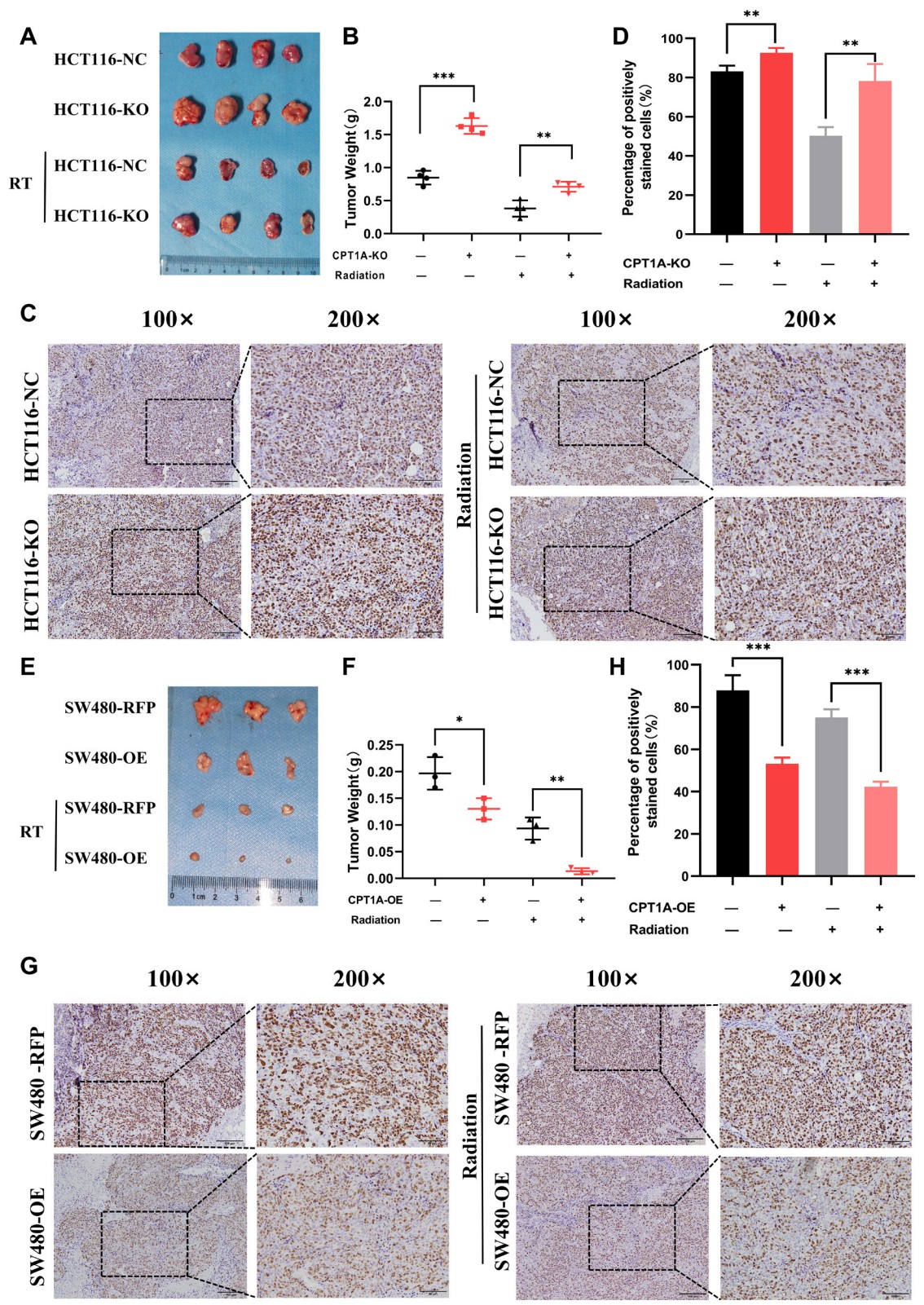

**Figure 4.** CPT1A inhibited proliferation and radioresistance in nude mice. (**A**) Image of tumours formed in nude mice, with knockout of CPT1A and radiation. (**B**) Scattergram showing the weight of tumours. (**C**) Immunohistochemical staining of Ki67 in tumours, scale bar = 100μm. . (**D**) Bar chart demonstrating the percentage of positively stained Ki67 cells. (**E**) Image of tumours formed in nude mice, with overexpression of CPT1A and radiation. (**F**) Scattergram showing the weight of tumours. (**G**) Immunohistochemical staining of Ki67 in tumours, scale bar = 100μm. (**H**) Bar chart demonstrating the percentage of positively stained Ki67 cells. ***p<0.001, **p<0.01, *p<0.05.

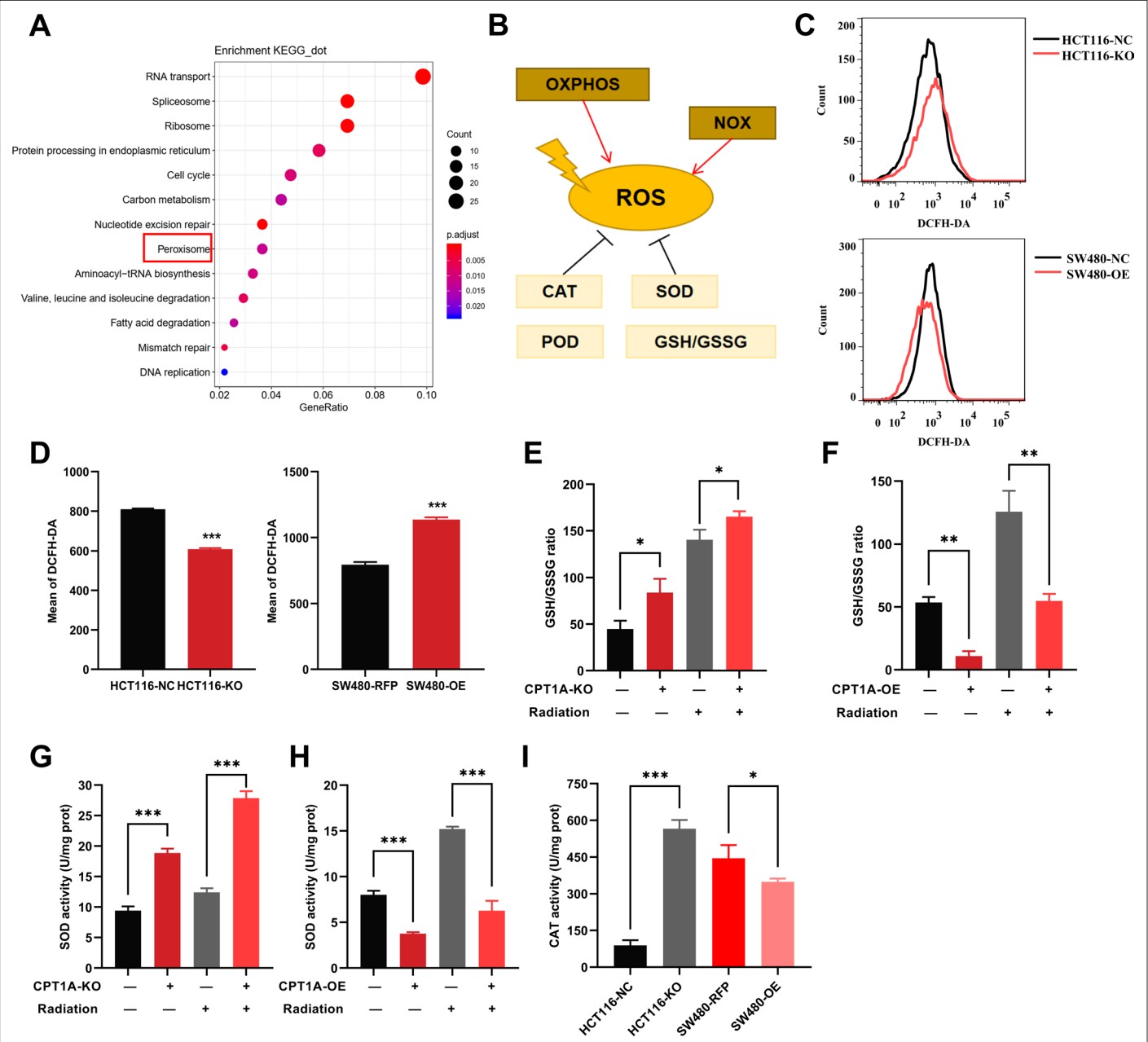

**Figure 5.** The effect of CPT1A on reactive oxygen species (ROS) related enzyme activity. (**A**) Enriched Kyoto Encyclopedia of Genes and Genomes (KEGG) pathway of differentially expressed genes (DEGs) in mRNA sequencing. (**B**) Generation and scavenging of ROS in cell. (**C**) ROS of HCT 116-KO cell, SW480-OE cell and their control with 2',7'-dichlorodihydrofluorescein diacetate (DCFH-DA) by flow cytometry. (**D**) Bar graph to show the mean of DCFH-DA in HCT 116-KO, HCT 116-NC cells, SW480-NC, and SW480-OE cells. (**E**) GSH/GSSG ratio measurement under CPT1A knockout and radiation. (**F**) GSH/GSSG ratio measurement under *CPT1A* overexpression and radiation. (**G**) Effect of the CPT1A knockout on SOD activity. (**H**) Effect of the *CPT1A* overexpression on SOD activity. (**I**) Effect of the CPT1A on catalase (CAT) activity. ***$p<0.001$, **$p<0.01$, *$p<0.05$.

The online version of this article includes the following figure supplement(s) for figure 5:

**Figure supplement 1.** The result of mRNA sequencing after CPT1A knockout.

**Figure supplement 2.** The effect of CPT1A on reactive oxygen species (ROS) and enzyme activity in radioresistance cell after 6 Gy radiation.

irradiation is through the X-ray ionisation of water molecules, which far exceeds those from oxidative phosphorylation and NOX enzymes (*Figure 5B*). Therefore, we speculated that the regulation of CPT1A by intracellular ROS levels may be attributed to increased ROS scavenging. We examined the GSH/GSSG ratio (*Figure 5E and F*) and SOD (*Figure 5G and H*), CAT (*Figure 5I*), and POD enzyme

activities (results were unchanged, data not shown) in stable *CPT1A* knockout/overexpression cells; in *CPT1A* knockout cells, the GSH/GSSG ratio and SOD and CAT enzyme activities increased, and these changes were also observed under 6 Gy irradiation (*Figure 5I*). In contrast, in *CPT1A*-overexpressing cells, the GSH/GSSG ratio, and SOD and CAT activities decreased, and these changes were also observed after 6 Gy irradiation (*Figure 5I*).

We further validated the effect of CPT1A on ROS scavenging in radioresistant cells. Total ROS significantly reduced in radioresistant cells compared to those in HCT-15 control cells following 6 Gy irradiation (*Figure 5—figure supplement 2A and C*). Furthermore, *CPT1A* overexpression increased total ROS levels in radioresistant cells (*Figure 5—figure supplement 2B and D*), suggesting that CPT1A enhances ROS accumulation in radioresistant cells. Additionally, we assessed the GSH/GSSG ratio (*Figure 5—figure supplement 2E and F*) and SOD (*Figure 5—figure supplement 2G and H*), POD (*Figure 5—figure supplement 2I and J*), and CAT enzyme activities (*Figure 5—figure supplement 2K*) in radioresistant cells, both at baseline and after 6 Gy irradiation, revealing an increased GSH/GSSG ratio and elevated activities of SOD, POD, and CAT enzymes in radioresistant cells compared to those in parental HCT-15 control cells. Furthermore, these changes persisted under 6 Gy irradiation.

To explore the influence of *CPT1A* overexpression on radiation resistance, we examined the GSH/GSSG ratio (*Figure 5—figure supplement 2E and F*) and SOD (*Figure 5—figure supplement 2G and H*), POD (*Figure 5—figure supplement 2I and J*), and CAT enzyme activities (*Figure 5—figure supplement 2K*) in HCT-15-25F-OE cells compared to those in control cells, showing that *CPT1A* overexpression restored the GSH/GSSG ratio and increased SOD and CAT enzyme activities but failed to restore POD enzyme activity. Additionally, under 6 Gy irradiation, the GSH/GSSG ratio and SOD enzyme activity were restored in CPT1A-overexpressing cells (*Figure 5—figure supplement 2E–K*).

## Reduced CPT1A expression mediates radioresistance in CRC through increased expression of ROS-scavenging genes, facilitated by FOXM1

The reasons underlying the changes in SOD and CAT enzyme activities require further investigation. We examined the transcriptional and protein levels of SOD (SOD1, SOD2, and SOD3) and CAT (*Figure 6A and B*) and found that *CPT1A* knockout in cells increased both the mRNA and protein levels of SOD1, SOD2, and CAT, whereas the overexpression of CPT1A decreased them (*Figure 6A and B*). However, the function of CPT1A as a transcription factor is hitherto unreported, suggesting a potential regulatory mechanism mediated by other transcription factors. Thus, we employed a bioinformatics analysis by intersecting predicted or reported transcription factors known to regulate SOD1, SOD2, and CAT with upregulated DEGs. The Venn diagram highlights three transcription factors that met the criteria: FOXM1, LMNB1, and SAP30 (*Figure 6C*). FOXM1 transcription and protein levels were significantly increased in *CPT1A* knockout cells but decreased when *CPT1A* was overexpressed (*Figure 6A and B*). Other transcription factors showed no significant changes in protein levels (data not shown).

Additionally, we explored the correlation between FOXM1 and the downstream enzymes SOD1, SOD2, and CAT in READ and COAD. The results (*Figure 6—figure supplement 1A and B*) indicate a positive correlation between FOXM1 and SOD1 as well as SOD2. Furthermore, we used hTFtarget and JASPAR to predict FOXM1 binding sites in the promoters of *SOD1*, *SOD2*, and *CAT* (*Table 1*). Finally, a rescue experiment was conducted by overexpressing *FOXM1* in HCT 116-NC and HCT 116-KO cells, demonstrating that downstream SOD1, SOD2, and CAT protein levels were restored by *FOXM1* overexpression (*Figure 6D*). In summary, the downregulation of *CPT1A* increases *FOXM1* mRNA and protein levels in CRC, promoting the transcription and translation of SOD1, SOD2, and CAT, thereby accelerating the scavenging of ROS produced after radiation exposure and ultimately leading to radiation resistance in CRC cells (*Figure 6E*).

## Discussion

We elucidated the role of CPT1A in CRC and the molecular mechanisms involved in mediating radiosensitivity. CPT1A is often downregulated in CRC, and low CPT1A expression can worsen OS and increase the probability of radiochemotherapy resistance. Low CPT1A expression increases FOXM1 activity, promoting the transcription and translation of downstream SOD1, SOD2, and CAT, thereby facilitating the scavenging of radiation-induced ROS. Our study establishes CPT1A as an effective

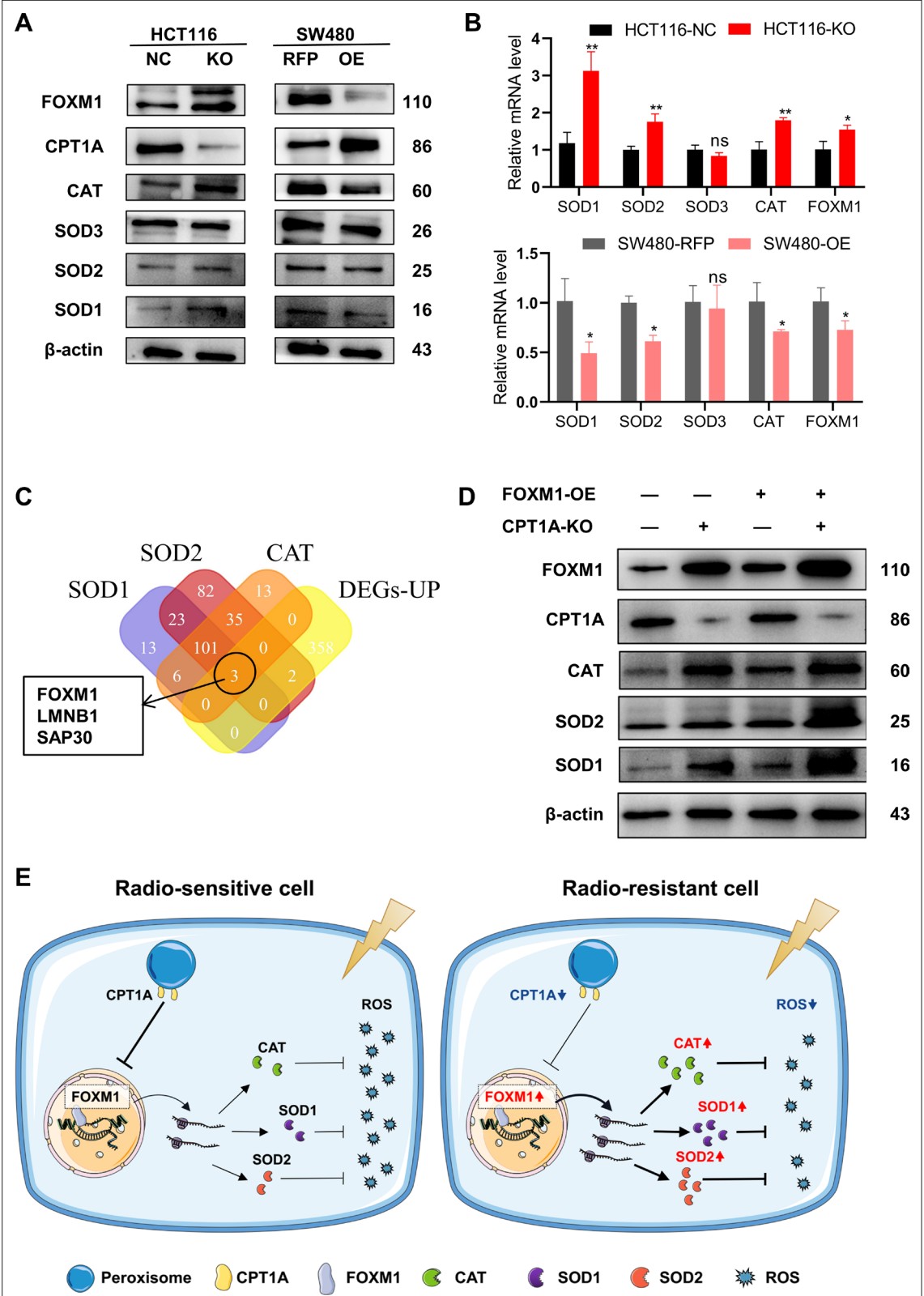

**Figure 6.** CPT1A increases the transcription and protein of reactive oxygen species (ROS) scavenge-related genes by regulating the transcription factor activity of FOXM1. (**A**) The protein level of FOXM1, CPT1A, catalase (CAT), SOD1, SOD2, SOD3 after knockout and overexpression of CPT1A. (**B**) The mRNA level of FOXM1, CAT, SOD1, SOD2, SOD3 after knockout and overexpression of CPT1A. (**C**) Venn diagram showing the potential transcription factor of SOD1, SOD2, and CAT. (**D**) The protein level of FOXM1, CPT1A, CAT, SOD1, SOD2 after overexpression of FOXM1 in HCT116-CPT1AKO cells.

*Figure 6 continued on next page*

*Figure 6 continued*

(**E**) Schematic diagram summarising our working model, namely, decreased CPT1A promotes the transcription factor activity of FOXM1, increasing the mRNA and protein level of CAT, SOD1, and SOD2, followed by increasing ROS scavenge after irradiation and therefore colorectal cancer (CRC) cells become radioresistance. ***p<0.001, **p<0.01, *p<0.05.

The online version of this article includes the following source data and figure supplement(s) for figure 6:

**Source data 1.** Original files for western blot analysis displayed in *Figure 6*.

**Source data 2.** PDF file containing original western blots for *Figure 6*.

**Figure supplement 1.** The correlation of FOXM1 with reactive oxygen species (ROS) scavenge gene.

biomarker to predict CRC prognosis and radiotherapy sensitivity and proposes the molecular mechanism by which it mediates radiosensitivity through the FOXM1-SOD1/SOD2/CAT axis.

CPT1A is crucial to CRC initiation and progression (*Mazzarelli et al., 2007*). However, its role in CRC remains unclear. Using a weighted gene co-expression network analysis to explore hub genes in CRC development, we found that CPT1A is expressed at low levels in CRC and acts as a central anti-cancer gene, exhibiting excellent prognostic value (*Wang et al., 2023*). Sinomenine improves colitis-associated cancer by upregulating CPT1A (*Zhang et al., 2022*). In contrast, high CPT1A expression is associated with malignancy in CRC, and its inhibition ameliorates malignant phenotypes. CPT1A-mediated fatty acid oxidation promotes CRC metastasis (*Wang et al., 2018*). DHP-B, a CPT1A inhibitor, disrupts CPT1A-VDAC1 interaction in the mitochondria, increasing mitochondrial permeability and reducing oxygen consumption and energy metabolism in CRC cells (*Hu et al., 2023*). We observed significant downregulation of CPT1A expression in CRC, and low CPT1A expression was associated with worse prognosis and greater radiochemotherapy resistance, contrary to previous reports. This discrepancy may be related to the inherent heterogeneity of tumour tissue and differences in tumour stage.

The PGC1α/CEBPB/CPT1A axis, which enhances lipid β-oxidation, increases ATP and NADPH levels and promotes cellular radiation resistance in nasopharyngeal carcinoma (*Tan et al., 2018*). Tan et al. also discovered an interaction between CPT1A and Rab14, which transports fatty acids into the mitochondria, thereby facilitating lipid oxidation and cell survival under irradiation (*Du et al., 2019*). The HER1/2-MEK-ERK1/2-CPT1A/CPT2 axis reportedly enhances cell proliferation and confers radiation resistance in breast cancer (*Han et al., 2019*). However, no studies have investigated the association between CPT1A and radiosensitivity in CRC. Our research revealed that CPT1A is a radiation sensitivity gene, contradicting previous literature, possibly due to differences in cancer types.

Transcriptomic sequencing revealed that CPT1A regulates multiple pathways, including the peroxisomal pathway, which is responsible for ROS scavenging. *CPT1A* knockdown upregulated FOXM1, which, in turn, stimulated the transcription and translation of crucial antioxidative enzymes, SOD1, SOD2, and CAT, thereby expediting ROS clearance. This contradicts previous findings that *CPT1A* overexpression accelerates ROS production by increasing fatty acid β-oxidation, thereby promoting ageing phenotypes or augmenting cancer cells' oxidative defences (*Jiang et al., 2022*; *Joshi et al., 2020*; *Luo et al., 2021*). Our findings diverge from those of other studies for two main reasons: first, previous research into the effects of CPT1A on ROS largely centred around mitochondria rather than the peroxisome; second, previous studies did not account for radiation, which significantly increases ROS production above cellular oxidative processes.

Forkhead box M1 (FOXM1) is a critical transcription factor for many cellular processes (*Kalathil et al., 2020*). Besides benefitting normal cell functions, it also regulates cancer processes, including growth, metastasis, and recurrence (*Alimardan et al., 2023*; *Khan et al., 2023*). FOXM1 also affects

**Table 1.** Potential binding site of SOD1, SOD2, catalase (CAT) promoter predicted by hTFtarget and JASPAR.

| TF | Target gene | Sequence name | Start | Stop | Strand | Score | p-Value | q-Value | Matched motif |
|---|---|---|---|---|---|---|---|---|---|
| FOXM1 | SOD1 | NC_000021.9:31657693–31659693 | 1166 | 1178 | - | 14.4412 | 0.00000152 | 0.00599 | TTTGTTTGATTTT |
| FOXM1 | SOD2 | NC_000006.12:c159669069 -159667069 | 1152 | 1160 | + | 12.685 | 0.00000403 | 0.0159 | AGATGGAGT |
| FOXM1 | CAT | NC_000011.10:34436934–34438934 | 1410 | 1422 | + | 11.3333 | 0.0000499 | 0.197 | TCAGAGTGTTTTT |

radiotherapy outcomes in many cancer types, including CRC (*Kwon et al., 2021*; *Li et al., 2022*; *Liu et al., 2019*; *Pal et al., 2018*; *Takeshita et al., 2023*; *Xiu et al., 2018*). In this study, we identified FOXM1 as a key regulator connecting CPT1A to ROS scavenging in CRC, exhibiting an inverse correlation with both CPT1A and ROS levels. FOXM1 is an essential transcription factor in intracellular redox, specifically in regulating the redox state of malignant mesothelioma cells (*Cunniff et al., 2014*). FOXM1-dependent fatty acid oxidation-mediated ROS modulation is a cell-intrinsic drug resistance mechanism in cancer stem cells (*Choi et al., 2020*). *FOXM1* knockdown increases intracellular ROS levels and decreases the transcription levels of SOD2, CAT, PRDX, and GPX2 (*Smirnov et al., 2016*). Overall, our findings align with existing literature, highlighting the crucial role of FOXM1 in orchestrating the interplay between CPT1A and ROS homoeostasis. We further revealed that FOXM1 participates in the transcriptional regulation of SOD1.

Our study has some limitations. We only conducted radiosensitivity investigations of CPT1A in nude mice, which only demonstrated its regulatory role in CRC cell radiosensitivity. It remains unclear whether CPT1A can regulate the radiosensitivity of the entire tumour microenvironment in immunocompetent mice. The cell lines used in our study have different genetic backgrounds; specifically, HCT116 is microsatellite instable, whereas SW480 is not. Additionally, FOXM1 primarily localises to the nucleus, whereas CPT1A is a cytoplasmic protein; there is no known physiological basis for their co-localisation. Therefore, the specific mechanism through which CPT1A regulates FOXM1 expression requires further investigation. The reasons for the decreased expression of CPT1A in tumour cells remain unclear. Future studies should explore this in greater detail.

## Materials and methods
### Reagents and materials
The reagent suppliers are indicated in Key resources table.

**Key resources table**

| Reagent type (species) or resource | Designation | Source or reference | Identifiers | Additional information |
|---|---|---|---|---|
| Cell line (human) | HCT-15 | ATCC (USA) | CCL-225 | |
| Cell line (human) | RKO | ATCC (USA) | CRL-2577 | |
| Cell line (human) | HCT 116 | ATCC (USA) | CCL-247 | |
| Cell line (human) | HT-29 | ATCC (USA) | HTB-38 | |
| Cell line (human) | Caco-2 | ATCC (USA) | HTB-37 | |
| Cell line (human) | SW480 | ATCC (USA) | CCL-228 | |
| Cell line (human) | SW620 | ATCC (USA) | CCL-227 | |
| Transfected construct (human) | pLV[Exp]-Puro-CMV>hCPT1A | VectorBuilder (China) | | |
| Transfected construct (human) | pLV[CRISPR]-hCas9:T2A:Puro-U6>hCPT1A | VectorBuilder (China) | | Including 3 gRNA |
| Transfected construct (human) | pLV[Exp]-G418-CMV>hFOXM1 | VectorBuilder (China) | | |
| Antibody | CPT1A (D3B3) Rabbit mAb | Cell Signaling Technology (USA) | 12252 | |
| Antibody | β-Actin (8H10D10) Mouse mAb | Cell Signaling Technology (USA) | 3770 | |
| Antibody | Mouse anti-Ki67 monoclonal antibody | Cell Signaling Technology (USA) | 9449 | |
| Antibody | PPARA Rabbit Polyclonal antibody | Proteintech (China) | 15540-1-AP | |
| Antibody | PPAR gamma Rabbit Polyclonal antibody | Proteintech (China) | 22061-1-AP | |
| Antibody | PGC1a Mouse Monoclonal antibody | Proteintech (China) | 66369-1-Ig | |

*Continued on next page*

*Continued*

| Reagent type (species) or resource | Designation | Source or reference | Identifiers | Additional information |
|---|---|---|---|---|
| Antibody | SOD1 Rabbit Polyclonal antibody | Proteintech (China) | 10269-1-AP | |
| Antibody | SOD2 Rabbit Polyclonal antibody | Proteintech (China) | 24127-1-AP | |
| Antibody | SOD3 Rabbit Polyclonal antibody | Proteintech (China) | 14316-1-AP | |
| Antibody | Catalase Rabbit Polyclonal antibody | Proteintech (China) | 21260-1-AP | |
| Antibody | FOXM1 Rabbit Polyclonal antibody | Proteintech (China) | 13147-1-AP | |
| Antibody | Anti-rabbit IgG, HRP-linked Antibody | Cell Signaling Technology (USA) | 7074 | |
| Chemical compound, drug | TRIzol | TakaraBio (Japan) | 9108 | |
| Commercial assay or kit | Evo M-MLV RT Mix Kit | Accurate Biotechnology (China) | AG11728 | |
| Commercial assay or kit | SYBR Green Premix Pro Taq HS qPCR Kit | Accurate Biotechnology (China) | AG11701 | |
| Commercial assay or kit | Protein extraction kit | KeyGen BioTech (China) | KGP113-SDS | |
| Commercial assay or kit | Comet assay kit | KeyGen BioTech (China) | KGA240 | |
| Chemical compound, drug | PI | Beyotime (China) | ST511 | |
| Chemical compound, drug | DAPI | Beyotime (China) | C1002 | |
| Commercial assay or kit | ROS detection kit | Beyotime (China) | S0033S | |
| Commercial assay or kit | GSH detection kit | Solarbio (China) | BC1175 | |
| Commercial assay or kit | GSSG detection kit | Solarbio (China) | BC1180 | |
| Commercial assay or kit | SOD enzyme activity kit | Solarbio (China) | BC0170 | |
| Commercial assay or kit | CAT enzyme activity kit | Solarbio (China) | BC0200 | |
| Commercial assay or kit | POD enzyme activity kit | Solarbio (China) | BC0090 | |

## Bioinformatic analyses

Raw mRNA expression profiles and clinical features from GSE9348, GSE20916, GSE37364, GSE44076, GSE68468, and GSE110223 were downloaded from GEO (http://www.ncbi.nlm.nih.gov/geo/). The *CPT1A* mRNA expression in cancer and paired normal tissues was analysed using UALCAN (https://ualcan.path.uab.edu/) (*Chandrashekar et al., 2022*). Correlation analyses between genes were conducted using GEPIA2 (http://gepia2.cancer-pku.cn/) (*Tang et al., 2019*). Transcription factor prediction and promoter binding site analysis for SOD1, SOD2, and CAT were conducted using the hTF target database (http://bioinfo.life.hust.edu.cn/hTFtarget#) (*Zhang et al., 2020*). Additionally, JASPAR (http://jaspar.genereg.net/) was used to predict the binding sites of FOXM1 (*Rauluseviciute et al., 2024*).

## Human

Human samples were collected from the Nanfang Hospital of Southern Medical University, Guangzhou, for CRC patients with or without radiochemotherapy with age over 18 and below 75 years. Two to four biopsies from each patient were collected to be used for qRT-PCR, western blotting, and sectioning. Informed consent forms were acquired from all the patients. Ethics approval for the use of human samples was obtained from Ethics Committee of Nanfang Hospital, Southern Medical University (No. NFEC-202304-K13).

**Table 2.** Correlation between clinicopathological features and the expression of CPT1A in tumour paraffin section.

| Variables | Categories | CPT1A | | | p-Value |
|---|---|---|---|---|---|
| | | Low | High | Total (n) | |
| Age | <50 | 13 | 13 | 26 | 0.411 |
| | ≥50 | 20 | 30 | 50 | |
| Gender | Male | 13 | 18 | 31 | 0.831 |
| | Female | 20 | 25 | 45 | |
| Histological grade | Well differentiated | 3 | 5 | 8 | 0.640 |
| | Moderately differentiated | 26 | 32 | 58 | |
| | Poorly differentiated | 4 | 6 | 10 | |
| UICC/ AJCC Stage | Stage I | 3 | 1 | 4 | 0.379 |
| | Stage II | 14 | 16 | 30 | |
| | Stage III | 14 | 24 | 38 | |
| | Stage IV | 2 | 2 | 4 | |
| T-class | T1 | 1 | 1 | 2 | 0.272 |
| | T2 | 3 | 3 | 6 | |
| | T3 | 15 | 15 | 30 | |
| | T4 | 14 | 24 | 38 | |
| N-class | N0 | 18 | 17 | 35 | 0.189 |
| | N1 | 10 | 16 | 26 | |
| | N2 | 5 | 10 | 15 | |
| M-class | M0 | 31 | 41 | 72 | 0.788 |
| | M1 | 2 | 2 | 4 | |

## RNA isolation and qRT-PCR

A total of 48 rectal adenocarcinoma samples and their paired normal tissues were used for RNA isolation and qRT-PCR. Total RNA was extracted using TRIzol following the manufacturer's instructions. cDNA was generated using the Evo M-MLV RT Mix Kit. mRNA expression was analysed using the SYBR Green Premix Pro Taq HS qPCR Kit on a QuantStudio6 Real-time PCR system, and β-actin was used for normalisation. Data were analysed using the $2^{-\Delta\Delta CT}$ method. The PCR primers are listed in *Supplementary file 2*.

## Protein extraction and western blotting

Proteins of rectal adenocarcinoma and paired normal tissues were extracted from tissues and cells using a protein extraction kit, according to the manufacturer's instructions. Briefly, NanoDrop was used to determine the protein concentration. The proteins were mixed in a 4:1 ratio with 5× loading buffer and denatured at 100°C in a water bath for 5 min. Subsequently, 20 μg of proteins were subjected to SDS-PAGE at 95 V. Subsequently, proteins were transferred onto PVDF membranes at a constant current. The membrane was blocked with 5% BSA and incubated with the corresponding primary antibody overnight at 4°C. Subsequently, the membrane was incubated with an HRP-conjugated secondary antibody (1:2000 dilution), and protein levels were detected using enhanced chemiluminescence.

## Immunohistochemistry

All colon cancer (n=76) and rectal adenocarcinoma tissues (n=45) were collected before treatment and sectioned at 4 μm. Clinicopathological features of the patients were provided by the Department of Pathology (*Table 2*). Staining intensity was independently evaluated by two senior pathologists.

The staining intensity was scored on a 4-point scale, where 0 represented no positive staining (negative), 1 represented light yellow (weak positive), 2 represented brownish yellow (positive), and 3 represented brown (strong positive) staining. The percentage of positive cells was also scored on a 4-point scale, where 1 point was assigned for ≤25% positive cells, 2 points for 26–50% positive cells, 3 points for 51–75% positive cells, and 4 points for >75% positive cells. The IHC score was calculated by multiplying the staining intensity score with the positive cell percentage score. Patients with an IHC score ≥6 were classified as the CPT1A-high group.

## Cell culture and lentiviral infection

The CRC cell lines HCT-15, RKO, HCT 116, HT-29, Caco-2, SW480, and SW620 were purchased from ATCC (Manassas, VA, USA). All cell lines were authenticated with STR profiling, using cell line authentication services offered by ATCC, to avoid misidentification and tested negative for mycoplasma contamination. All cell lines were cultured in RPMI-1640 medium supplemented with 10% FBS. Cells were cultured at 37°C with 5% $CO_2$. To establish radiation-resistant strains, HCT-15-25F and HCT-15-5F cells were generated using both conventional fractionated irradiation (2 Gy/fraction, 25 fractions, 5 fractions/week over 5 weeks) and large-fractionated irradiation (5 Gy/fraction, 5 fractions, for 1 week).

The full-length lentiviral expression of *CPT1A* with puromycin was constructed by VectorBuilder. *CPT1A*-targeting CRISPR/Cas9 lentiviral vectors (hCpt1a[gRNA#1]: AAATCTCTACTACACGGCCG ATGTTACGACAGGTACCGTCCTT, hCpt1a[gRNA#2]: AGAAGGTAAGGACGGTACCTGTCGTAAC ATCGGCCGTGTAGTA, hCpt1a[gRNA#3]: CTGAACACTCCTGGGCAGATGCGCCGATCGTGGCCC ACCTTTG) with RFP and Puro were constructed by VectorBuilder. Infection and in vitro transfection of cell lines were performed following the manufacturer's protocol. The lentiviral full-length expression of *FOXM1* with G418 was constructed using VectorBuilder.

## CFA and multi-target single-hit survival model

The radiosensitivity of all human CRC cell lines was determined using a CFA with a multi-target single-hit model to the surviving fractions. Cells were plated in six-well plates and irradiated at doses of 0, 2, 4, 6, 8, and 10 Gy (6 MeV X-rays). The cells were then cultured for 10 days, stained with 1% crystal violet, and quantified using ImageJ version 1.8.0.

The surviving fraction for each dose was calculated using the following formula:

[(number of surviving colonies at dose X)/(number of cells seeded at dose X (average colonies arising from non-irradiated cells (0 Gy))/number of non-irradiated cells seeded)]. Survival curves were used to develop the multi-target single-hit model, $SF = 1 - (1 - e^{-D/D_0}) \times N$, where SF is the surviving fraction, D is the radiation dose, and N is the extrapolation number. The multi-target single-hit model provides parameters related to radiation sensitivity and sensitisation ratios, thereby directly reflecting the cells' radiation sensitivity.

## Comet assay

The comet assay is a classical technique for assessing cellular radiation sensitivity. Essentially, cells that are more susceptible to radiation damage exhibit more DNA fragmentation post-irradiation, leading to a greater proportion of DNA in the tail (with less in the head). The comet assay was performed as previously described (*Yu et al., 2022*). Briefly, slides were covered with 100 μL of pre-warmed normal melting point agarose (2%) and placed on ice to solidify the first gel layer. Cells irradiated with 6 Gy were digested to obtain a cell suspension. Ten microlitres of the cell suspension were mixed with 80 μL of pre-warmed low melting point agarose (0.75%) and poured onto the slides. The slides were dipped in a cold lysis solution for 2 hr. After cell lysis, the slides were placed in a horizontal electrophoresis chamber filled with cold TAE solution, incubated for 25 min in the dark, and electrophoresed (1 V/cm). The slides were then neutralised in PBS for 5 min and stained with propidium iodide (PI) or DAPI. Comet images were captured using a fluorescence microscope. The percentage of DNA in the tail was analysed using the CASP 1.2.3 beta 1.

## Animal model

All animal experiments were approved by the Institutional Animal Care and Use Committee of Nanfang Hospital (IACUC-PROJECT-20221128-003). Male BALB/c nude mice (5 weeks of age) were purchased from the Southern Medical University Laboratory Animal Center, China, and raised under specific

pathogen-free conditions. All in vivo experiments were performed following institutional guidelines. To develop the xenograft tumour model, $5 \times 10^6$ cells were subcutaneously injected into the left flank of mice. On reaching a volume of 100 mm$^3$, the tumours were irradiated twice at 8 Gy, and other parts of the mouse body were protected with a lead shield. The tumour volume was measured using Vernier callipers and calculated as 1/2×length×width×width. After the mice were euthanised with phenobarbital sodium, the tumours were excised, weighed, and embedded in paraffin for further experiments.

## Transcriptomics

HCT 116-NC and HCT 116-KO cells were collected and sent to RIBOBIO (Guangzhou, China) for polyA-seq transcriptome sequencing. RNA extraction, library preparation, and sequencing were performed according to the manufacturer's instructions. To identify the DEGs between the two groups, the expression level of each transcript was calculated according to the transcript per million reads method. RSEM was used to quantify the gene abundance. Differential expression analysis was performed using DESeq2 (*Liu et al., 2021*). Genes with |log$_2$(fold change)|>1 and q<0.05 were considered significantly differentially expressed. In addition, functional enrichment analysis, including using GO and KEGG, was performed to identify significantly enriched DEGs in GO terms and metabolic pathways at a Bonferroni-corrected p-value of 0.05 compared with the whole-transcriptome background. The heatmap and volcano of mRNA sequencing were conducted on ImageGP (https://www.bic.ac.cn/ImageGP) (*Chen et al., 2022*). GO functional enrichment and KEGG pathway analyses were performed using the ClusterProfiler package in R (*Wu et al., 2021*).

## ROS detection

ROS were stained with a DCFH-DA probe and detected using flow cytometry. Briefly, the probe was diluted in cell culture media at a 1:1000 ratio to yield a final concentration of 10 µmol/L. The culture medium was replaced with 1 mL of DCFH-DA solution. The cells were incubated at 37°C for 20 min. After incubation, the cells were washed thrice with serum-free medium to remove any excess DCFH-DA.

One hour after 6 Gy irradiation, the cells were digested into single-cell suspensions and washed once with PBS. Finally, cells were resuspended in PBS, and the fluorescence intensity was analysed using flow cytometry at excitation and emission wavelengths of 488 and 525 nm, respectively.

## GSH/oxidised GSH ratio

Cells ($5 \times 10^5$ cells/well) were seeded into 10 cm wells until they reached 80% confluence. One hour after 6 Gy irradiation, the GSH/oxidised GSH (GSSG) ratio was determined using a GSH/GSSG ratio detection assay kit following the manufacturer's protocols.

## Enzyme activity

Cells ($5 \times 10^5$ cells/well) were seeded into 10 cm wells until reaching 80% confluence before. One hour after irradiation with 6 Gy, crude enzyme extract was prepared. The cells were digested, collected, and centrifuged to remove the supernatant. Next, 1 mL of HPLC-grade water was added to each pellet. The cells were disrupted by ultrasonication (20% power, 3 s on and 10 s off, repeated 25–40 times). The resulting mixture was centrifuged at 8000×$g$ and 4°C for 10 min, and the supernatant was collected as the crude enzyme extract. The enzyme activities of SOD1, SOD2, CAT, and POD were detected using respective enzyme activity kits, according to the manufacturer's protocols.

## Statistical analysis

Data are expressed as the mean ± standard deviation (SD), and p-values<0.05 were considered statistically significant in all experiments. Data were analysed using one-way analysis of variance (ANOVA), Spearman's correlation, and Kaplan-Meier estimates. Statistical analyses were performed using SPSS software (version 20.0). All experiments were performed in triplicate.

## Acknowledgements

We would like to thank Editage (https://www.editage.cn) for English language editing. This research project was supported by the National Natural Science Foundation of China (No. 82273564,

32370139, 32300085, and 32070118), Key Science & Technology Brainstorm Project of Guangzhou (No. 202206010045).

## Additional information

### Funding

| Funder | Grant reference number | Author |
|---|---|---|
| National Natural Science Foundation of China | 82273564 | Yi Ding |
| National Natural Science Foundation of China | 32370139 | Hongying Fan |
| National Natural Science Foundation of China | 32300085 | Zhenhui Chen |
| National Natural Science Foundation of China | 32070118 | Hongying Fan |
| Key Science and Technology Brainstorm Project of Guangzhou | 202206010045 | Yi Ding |

The funders had no role in study design, data collection and interpretation, or the decision to submit the work for publication.

### Author contributions

Zhenhui Chen, Data curation, Funding acquisition; Lu Yu, Data curation, Writing – original draft; Zhihao Zheng, Qiqing Guo, Yuqin Zhang, Methodology; Xusheng Wang, Data curation; Yuchuan Chen, Validation; Yaowei Zhang, Funding acquisition, Writing – review and editing; Jianbiao Xiao, Supervision; Keli Chen, Supervision, Writing – review and editing; Hongying Fan, Resources, Writing – review and editing; Yi Ding, Supervision, Funding acquisition

### Author ORCIDs

Lu Yu ⓘ https://orcid.org/0000-0003-4073-3842
Yi Ding ⓘ https://orcid.org/0000-0001-5035-9255

### Ethics

Human samples were collected from the Nanfang Hospital of Southern Medical University, Guangzhou, for colorectal cancer patients with or without radiochemotherapy with age over 18 and below 75 year. Two to four biopsies from each patient were collected to be used for qRT-PCR, western blotting, and sectioning. Informed consent forms were acquired from all the patients. Ethics approval for the use of human samples was obtained from Ethics Committee of Nanfang Hospital, Southern Medical University (No. NFEC-202304-K13).

All animal experiments were approved by the Institutional Animal Care and Use Committee of Nanfang Hospital (IACUC-PROJECT-20221128-003).

Reviewer #1 (Public review): https://doi.org/10.7554/eLife.97827.3.sa1
Reviewer #2 (Public review): https://doi.org/10.7554/eLife.97827.3.sa2
Author response https://doi.org/10.7554/eLife.97827.3.sa3

## Additional files

### Supplementary files

- Supplementary file 1. The $\overline{D}0$ and N value of multi-target single-hit model in all cell lines.
- Supplementary file 2. Primers used for the real-time PCR.
- MDAR checklist

## Data availability

Raw mRNA expression profiles and clinical features of the GSE9348, GSE20916, GSE37364, GSE44076, GSE68468 and GSE110223 datasets are available in the GEO database (http://www.ncbi.nlm.nih.gov/geo/). The raw mRNA expression profiles of rectal and colon cancer patients in the TCGA database are available ualcan database (https://ualcan.path.uab.edu/). The mRNA sequencing data supporting the findings of this study are available in the China National Center for Bioinformation (CNCB) under accession number PRJCA030073.

The following dataset was generated:

| Author(s) | Year | Dataset title | Dataset URL | Database and Identifier |
|---|---|---|---|---|
| Yu Lu | 2024 | CPT1A Mediates Radiation Sensitivity in Colorectal Cancer | https://ngdc.cncb.ac.cn/bioproject/browse/PRJCA030073 | NCBI BioProject, PRJCA030073 |

The following previously published datasets were used:

| Author(s) | Year | Dataset title | Dataset URL | Database and Identifier |
|---|---|---|---|---|
| Yi H | 2010 | Expression data from healthy controls and early stage CRC patient's tumor | https://www.ncbi.nlm.nih.gov/geo/query/acc.cgi?acc=GSE9348 | NCBI Gene Expression Omnibus, GSE9348 |
| Skrzypczak M, Goryca K, Rubel T, Paziewska A, Mikula M, Jarosz D, Pachlewski J, Oledzki J, Ostrowski J | 2010 | Modeling oncogenic signaling in colon tumors by multidirectional analyses of microarray data | https://www.ncbi.nlm.nih.gov/geo/query/acc.cgi?acc=GSE20916 | NCBI Gene Expression Omnibus, GSE20916 |
| Galamb O, Wichmann B, Molnar B | 2013 | Expression data from human colonic biopsy samples (adenoma-carcinoma) | https://www.ncbi.nlm.nih.gov/geo/query/acc.cgi?acc=GSE37364 | NCBI Gene Expression Omnibus, GSE37364 |
| Solé X, Crous-Bou M, Sanz-Pamplona R, Paré L, Cordero D, Guinó E, Berenguer A, Closa A, Olivares D, Lopez-Doriga A, Moreno V | 2014 | Gene expression data from healthy, adjacent normal and tumor colon cells | https://www.ncbi.nlm.nih.gov/geo/query/acc.cgi?acc=GSE44076 | NCBI Gene Expression Omnibus, GSE44076 |
| Heiskanen M | 2015 | caArray_notte-00422: Molecular Dissection of Colon Cancer | https://www.ncbi.nlm.nih.gov/geo/query/acc.cgi?acc=GSE68468 | NCBI Gene Expression Omnibus, GSE68468 |
| Vlachavas E, Papadodima O, Pilalis E, Koczan D, Willis S, Klippel S, Dimitrakopoulou-Strauss A, Pan L, Sachpekidis C, Chatziioannou A | 2018 | Expression data from 13 patients with colorectal cancer | https://www.ncbi.nlm.nih.gov/geo/query/acc.cgi?acc=GSE110223 | NCBI Gene Expression Omnibus, GSE110223 |

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
