## [Editor Report · eLife Assessment]

This study reports a **valuable** finding for the treatment of colorectal cancer (CRC), as the authors demonstrated that the enzyme CPT1A plays an significant role in the response to radiotherapy in CRC patients. However, the reviewers found that the results presented are still **incomplete**.

---

## [Referee Report · Reviewer #1 (Public review)]

Summary:

Fats and lipids serve many important roles in cancers, including serving as important fuels for energy metabolism in cancer cells by being oxidized in the mitochondria. The process of fatty acid oxidation is initiated by the enzyme carnitine palmitoyltransferase 1A (CPT1A), and the function and targetability of CPT1A in cancer metabolism and biology has been heavily investigated. This includes studies that have found important roles for CPT1A in colorectal cancer growth and metastasis.

In this study, Chen and colleagues use analysis of patient samples and functional interrogation in animal models to examine the role CPT1A plays in colorectal cancer (CRC). The authors find that CPT1A expression is decreased in CRC compared to paired healthy tissue and that lower expression correlates with decreased patient survival over time, suggesting that CPT1A may suppress tumor progression. To functionally interrogate this hypothesis, the authors both use CRISPR to knockout CPT1A in a CRC cell line that expresses CPT1A, and overexpress CPT1A in a CRC cell line with low expression. In both systems, increased CPT1A expression decreased cell survival and DNA repair in response to radiation in culture. Further, in xenograft models CPT1A decreased tumor growth basally and radiotherapy could further decrease tumor growth in CPT1A expressing tumors. As CRC is often treated with radiotherapy, the authors argue this radiosensitization driven by CPT1A could explain why CPT1A expression correlates with increased patient survival.

Lastly, Chen and colleagues sought to understand why CPT1A suppresses CRC tumor growth and sensitizes the tumors to radiotherapy in culture. Antioxidant capacity of cells can increase cell survival, so the authors examine antioxidant gene expression and levels in CPT1A expressing and non-expressing cells. CPT1A expression suppresses expression of antioxidant metabolism genes and lowers levels of antioxidants. Antioxidant metabolism genes can be regulated by the FOXM1 transcription factor, and the authors find that CPT1A expression regulates FOXM1 levels and that antioxidant gene expression can be partially rescued in CPT1A expressing CRC cells. This leads the authors to propose the following model: CPT1A expression downregulates FOXM1 (via some yet undescribed mechanism) which then leads to decreased antioxidant capacity in CRC cells and thus suppressing tumor progression and increasing radiosensitivity. This is an interesting model that could explain suppression of CPT1A expression in CRC, but key tenets of the model are untested and speculative.

Strengths:

• Analysis of CPT1A in paired CRC tumors and non-tumor tissue using multiple modalities combined with analysis of independent datasets rigorously show that CPT1A is downregulated in CRC tumors at the RNA and protein level.

• The authors use paired cell line model systems where CPT1A is both knocked out and overexpressed in cells lines that endogenously express or repress CPT1A respectively. These complementary model systems increase the rigor of the study.

• The finding that a metabolic enzyme generally thought to support tumor energetics actually is a tumor suppressor in some settings is theoretically quite interesting.

Weaknesses:

• The authors propose that CPT1A expression modulates antioxidant capacity in cells by suppressing FOXM1 and that this pathway alters CRC growth and radiotherapy response. However, key aspects of this model are not tested. The authors do not show that FOXM1 contributes to regulation of antioxidant levels in CRC cells and tumors or if FOXM1 suppression is key to inhibition of CRC tumor growth and radiosensitization by CPT1A. Thus, the model the authors propose is speculative and not supported by the existing data.

• The authors propose two mechanisms by which CPT1A expression triggers radiosensitization: decreasing DNA repair capacity (Fig. 3) and decreasing antioxidant capacity (Fig. 5). However, while CPT1A expression does alter these capacities in CRC cells, neither is functionally tested to determine if altered DNA repair or antioxidant capacity (or both) are the reason why CRC cells are more sensitive to radiotherapy or are delayed in causing tumors in vivo. Thus, this aspect of the proposed model is also speculative.

• The authors find that CPT1A affects radiosensitization in cell culture and assess this in vivo. In vivo, CPT1A expression slows tumor growth even in the absence of radiotherapy, and radiotherapy only proportionally decreases tumor growth to the same extent as it does in CPT1A non-expressing CRC tumors. The authors propose from this data that CPT1A expression also sensitizes tumors to radiotherapy in vivo. However, it is unclear that CPT1A expression causes radiosensitization in vivo or if CPT1A expression acts as independent tumor suppressor to which radiotherapy has an additive effect. Additional experiments would be necessary to differentiate between these possibilities.

• The authors propose in Figure 3 that DNA repair capacity is inhibited in CRC cells by CPT1A expression. However, the gH2AX immunoblots performed in Figure 3H-I that measure DNA repair kinetics are not convincing that CPT1A expression impairs DNA repair kinetics. Separate blots are shown for CPT1A expressing and non-expressing cell lines, not allowing for rigorous comparison of gH2AX levels and resolution as CPT1A expression is modulated.

---

## [Referee Report · Reviewer #2 (Public review)]

The manuscript by Chen et al. describes how low levels of CPT1A in colorectal cancer (CRC) confer radioresistance by expediting radiation-induced ROS clearance. The authors propose that this mechanism of ROS homeostasis is regulated through FOXM1. CPT1A is known for its role in fatty acid metabolism via beta-oxidation of long-chain fatty acids, making it important in many metabolic disorders and cancers.

Previous studies have suggested that upregulation of CPT1A is essential for the tumor-promoting effect in colorectal cancers (CRC) (PMID: 32913185). For example, CPT1A-mediated fatty acid oxidation promotes colorectal cancer cell metastasis (PMID: 29995871), and repression of CPT1A activity renders cancer cells more susceptible to killing by cytotoxic T lymphocytes (PMID: 37722058). Additionally, CPT1A-mediated fatty acid oxidation (FAO) sensitizes nasopharyngeal carcinomas to radiation therapy (PMID: 29721083). While this suggests a tumor-promoting effect for CPT1A, the work by Chen et al. suggests instead a tumor-suppressive function for CPT1A in CRC, specifically that loss or low expression of CPT1A confers radioresistance in CRC. This makes the findings important given that they oppose the previously proposed tumorigenic function of CPT1A.

The study has several strengths. The authors employ both in vitro and in vivo models to demonstrate that low CPT1A levels lead to radioresistance in CRC cells. They use isogenic HCT15 CRC cell lines that are radioresistant and show that overexpression of CPT1A sensitizes these cells to radiotherapy. Interestingly, the radioresistant cells exhibit lower CPT1A levels, suggesting that downregulation of CPT1A may be involved in the acquisition of radioresistance. Throughout the manuscript, the authors acknowledge the limitations of their work and avoid overextending their conclusions.

However, there are some major limitations to the study:

(1) Unexplored Contradictions with Previous Studies

While the authors propose a tumor-suppressive function for CPT1A in CRC, they do not sufficiently address the contradiction with prior studies that indicate a tumor-promoting role for CPT1A. The discussion briefly mentions that this discrepancy may stem from heterogeneity or differences in tumor stages, but a more thorough exploration is needed. Delving deeper into the contexts and conditions under which CPT1A exhibits differing roles would be critical for reconciling these findings and guiding future research.

(2) Limited Patient Data Analysis

The authors demonstrate that CPT1A levels are significantly lower in COAD (colon adenocarcinoma) and READ (rectal adenocarcinoma) compared to normal tissues. However, data from TCGA indicate that CPT1A expression levels are lower in 26 out of 31 tumor types compared to COAD or READ (as noted in the authors' response to the previous review). It is possible that reduced CPT1A expression might be a common feature across various cancers, not just CRC. A more comprehensive analysis comparing matched normal and tumor tissues across different cancer types would clarify whether the observed phenomenon is unique to CRC or part of a broader pattern. This is particularly important since several studies have reported CPT1A overexpression in tumors.

(3) Limitations in Experimental Scope

The experimental design primarily involves CPT1A knockout in HCT116 cells and CPT1A overexpression in SW480 cells, which may limit the generalizability of the findings. Utilizing additional cell lines would account for genetic heterogeneity and enhance the robustness of the conclusions. Moreover, while the authors suggest an opposing effect of CPT1A in CRC compared to other studies, they have not investigated this through pharmacological means. Previous studies have shown that pharmacological inhibition of CPT1A can limit cancer progression (e.g., PMID: 33528867, PMID: 32198139) and sensitize cells to radiation therapy (PMID: 30175155). Testing whether pharmacological inhibitors like etomoxir or ST1326 replicate the effects observed with genetic knockout would provide valuable insights and have significant implications for therapeutic strategies in CRC patients.

Conclusion

This study offers valuable insights into the role of CPT1A in CRC radioresistance, proposing a tumor-suppressive function that challenges previous findings of its tumor-promoting role. While the findings are interesting and could have significant implications for cancer therapy, the limitations in experimental scope and the lack of a thorough discussion reconciling contradictory evidence warrant caution. Expanding the research to include a wider range of CRC cell lines, conducting pharmacological inhibition studies, and performing more detailed analyses would strengthen the conclusions and enhance our understanding of CPT1A's complex role in cancer progression and treatment response.

---

## [Author Response]

The following is the authors’ response to the original reviews.

**Public Reviews:**

**Reviewer #1 (Public Review):**
Summary:Fats and lipids serve many important roles in cancers, including serving as important fuels for energy metabolism in cancer cells by being oxidized in the mitochondria. The process of fatty acid oxidation is initiated by the enzyme carnitine palmitoyltransferase 1A (CPT1A), and the function and targetability of CPT1A in cancer metabolism and biology have been heavily investigated. This includes studies that have found important roles for CPT1A in colorectal cancer growth and metastasis.In this study, Chen and colleagues use analysis of patient samples and functional interrogation in animal models to examine the role CPT1A plays in colorectal cancer (CRC). The authors find that CPT1A expression is decreased in CRC compared to paired healthy tissue and that lower expression correlates with decreased patient survival over time, suggesting that CPT1A may suppress tumor progression. To functionally interrogate this hypothesis, the authors both use CRISPR to knockout CPT1A in a CRC cell line that expresses CPT1A and overexpress CPT1A in a CRC cell line with low expression. In both systems, increased CPT1A expression decreased cell survival and DNA repair in response to radiation in culture. Further, in xenograft models, CPT1A decreased tumor growth basally and radiotherapy could further decrease tumor growth in CPT1A-expressing tumors. As CRC is often treated with radiotherapy, the authors argue this radiosensitization driven by CPT1A could explain why CPT1A expression correlates with increased patient survival.Lastly, Chen and colleagues sought to understand why CPT1A suppresses CRC tumor growth and sensitizes the tumors to radiotherapy in culture. The antioxidant capacity of cells can increase cell survival, so the authors examine antioxidant gene expression and levels in CPT1A-expressing and non-expressing cells. CPT1A expression suppresses the expression of antioxidant metabolism genes and lowers levels of antioxidants. Antioxidant metabolism genes can be regulated by the FOXM1 transcription factor, and the authors find that CPT1A expression regulates FOXM1 levels and that antioxidant gene expression can be partially rescued in CPT1A-expressing CRC cells. This leads the authors to propose the following model: CPT1A expression downregulates FOXM1 (via some yet undescribed mechanism) which then leads to decreased antioxidant capacity in CRC cells, thus suppressing tumor progression and increasing radiosensitivity. This is an interesting model that could explain the suppression of CPT1A expression in CRC, but key tenets of the model are untested and speculative.Strengths:Analysis of CPT1A in paired CRC tumors and non-tumor tissue using multiple modalities combined with analysis of independent datasets rigorously show that CPT1A is downregulated in CRC tumors at the RNA and protein level.The authors use paired cell line model systems where CPT1A is both knocked out and overexpressed in cell lines that endogenously express or repress CPT1A respectively. These complementary model systems increase the rigor of the study.The finding that a metabolic enzyme generally thought to support tumor energetics actually is a tumor suppressor in some settings is theoretically quite interesting.

We would like to thank Reviewer #1 for the positive comments.

Weaknesses:The authors propose that CPT1A expression modulates antioxidant capacity in cells by suppressing FOXM1 and that this pathway alters CRC growth and radiotherapy response. However, key aspects of this model are not tested. The authors do not show that FOXM1 contributes to the regulation of antioxidant levels in CRC cells and tumors or if FOXM1 suppression is key to the inhibition of CRC tumor growth and radiosensitization by CPT1A. Thus, the model the authors propose is speculative and not supported by the existing data.

We thank the reviewer for the valuable comment. In this study, we employed Western blotting to assess the protein levels of the ROS scavenging enzymes CAT, SOD1, and SOD2 following FOXM1 overexpression. This approach allowed us to evaluate how FOXM1 regulates ROS clearance and mediates cellular radiation resistance. Further *in-vivo* evidence is needed and will be addressed in future research.

The authors propose two mechanisms by which CPT1A expression triggers radiosensitization: decreasing DNA repair capacity (Figure 3) and decreasing antioxidant capacity (Figure 5). However, while CPT1A expression does alter these capacities in CRC cells, neither is functionally tested to determine if altered DNA repair or antioxidant capacity (or both) are the reason why CRC cells are more sensitive to radiotherapy or are delayed in causing tumors in vivo. Thus, this aspect of the proposed model is also speculative.

We thank the reviewer for the valuable comment. In this study, we combined a colony formation assay, multi-target single-hit survival model, comet assay, and Western blotting (for γH2AX) to evaluate DNA damage and repair in cells. Additionally, we employed qPCR, Western blotting, and enzyme activity kits to assess the direct ROS-scavenging activities of the peroxisomal enzymes CAT, SOD1, SOD2, and SOD3.

The authors find that CPT1A affects radiosensitization in cell culture and assess this in vivo. In vivo, CPT1A expression slows tumor growth even in the absence of radiotherapy, and radiotherapy only proportionally decreases tumor growth to the same extent as it does in CPT1A non-expressing CRC tumors. The authors propose from this data that CPT1A expression also sensitizes tumors to radiotherapy in vivo. However, it is unclear whether CPT1A expression causes radiosensitization in vivo or if CPT1A expression acts as an independent tumor suppressor to which radiotherapy has an additive effect. Additional experiments would be necessary to differentiate between these possibilities.

We thank the reviewer for the valuable comment. As shown in Figure 4D, in the absence of CPT1A knockdown, radiotherapy reduced the percentage of Ki67-positive cells in the xenograft tumors by 32.9% (approximately 39.6% of the pre-irradiation baseline). In contrast, upon CPT1A knockdown, radiotherapy only led to a 14.5% reduction in the percentage of Ki67-positive cells (approximately 15.6% of the pre-irradiation baseline). Furthermore, as illustrated in Figures 4E and 4F, in the absence of CPT1A overexpression, radiotherapy resulted in a 0.10-g decrease in tumor weight (around 52.5% of the pre-irradiation weight), whereas with CPT1A overexpression, radiotherapy induced a more pronounced 0.12-g reduction in tumor weight (approximately 89.7% of the pre-irradiation weight). Collectively, these findings indicate that CPT1A exhibits a radiosensitising effect. We have incorporated these relevant details in the Results section (Lines 196-201 and 204-208).

The authors propose in Figure 3 that DNA repair capacity is inhibited in CRC cells by CPT1A expression. However, the gH2AX immunoblots performed in Figure 3H-I that measure DNA repair kinetics are not convincing that CPT1A expression impairs DNA repair kinetics. Separate blots are shown for CPT1A expressing and non-expressing cell lines, not allowing for rigorous comparison of gH2AX levels and resolution as CPT1A expression is modulated.

We thank the reviewer for the valuable comment. In this study, we also employed a colony formation assay, multi-target single-hit survival model, and comet assay to elucidate the impact of CPT1A on DNA repair capacity. These methods all indicated that DNA repair capacity is inhibited in CRC cells by CPT1A expression.

There are conflicting studies (PMID: 37977042, 29995871) that suggest that CPT1A is overexpressed in CRC and contributes to tumor progression rather than acting as a tumor suppressor as the authors propose. It would be helpful for readers for the authors to discuss these studies and why there is a discrepancy between them.

We thank the reviewer for the valuable comment. We have expanded the discussion of these findings in the relevant section of the manuscript (Lines 317-318). We speculated that the differences between our observations and previous reports may be attributable to the inherent heterogeneity of tumor tissues as well as variations in tumor stage.

**Reviewer #2 (Public Review):**
The manuscript by Chen et al. describes how low levels of CPT1A in colorectal cancer (CRC) confer radioresistance by expediting radiation-induced ROS clearance. The authors propose that this mechanism of ROS homeostasis is regulated through FOXM1. CPT1A is known for its role in fatty acid metabolism via beta-oxidation of long-chain fatty acids, making it important in many metabolic disorders and cancers.Previous studies have suggested that the upregulation of CPT1A is essential for the tumor-promoting effect in colorectal cancers (CRC) (PMID: 32913185). For example, CPT1A-mediated fatty acid oxidation promotes colorectal cancer cell metastasis (PMID: 2999587), and repression of CPT1A activity renders cancer cells more susceptible to killing by cytotoxic T lymphocytes (PMID: 37722058). Additionally, inhibition of CPT1A-mediated fatty-acid oxidation (FAO) sensitizes nasopharyngeal carcinomas to radiation therapy (PMID: 29721083). While this suggests a tumor-promoting effect for CPT1A, the work by Chen et al. suggests instead a tumor-suppressive function for CPT1A in CRC, specifically that loss or low expression of CPT1A confers radioresistance in CRC. This makes the findings important given that they oppose the previously proposed tumorigenic function of CPT1A. However, the data presented in the manuscript is limited in scope and analysis.Major Limitations:(1) Analysis of Patient Samples- Figure 1D shows that CPT1A levels are significantly lower in COAD and READ compared to normal tissues. It would be beneficial to show whether CPT1A levels are also significantly lower in CRC compared to other tumor types using TCGA data.

We thank the reviewer for the valuable comment. We assessed the expression levels of CPT1A across all cancer types in the TCGA dataset and found that the abundance of CPT1A in CRC was significantly lower compared to cholangiocarcinoma (CHOL), esophageal carcinoma (ESCA), kidney chromophobe (KICH), acute myeloid leukemia (LAML), and stomach adenocarcinoma (STAD) (Author response image 1).

**Author response image 1. sa3fig1:** The mRNA level of CPT1A across all cancer types in the TCGA dataset.

- The analysis should include a comparison of closely related CPT1 isoforms (CPT1B and CPT1C) to emphasize the specific importance of CPT1A silencing in CRC.

We thank the reviewer for the valuable comment. We further examined the mRNA expression levels of the CPT1 isoforms CPT1B and CPT1C in COAD and READ tumor samples and their respective normal tissue counterparts. The results showed that CPT1B was significantly upregulated in READ tumor samples compared to normal tissues. Similarly, CPT1C was significantly overexpressed in both READ and COAD tumor samples relative to their normal tissue controls (Author response image 2).

**Author response image 2. sa3fig2:** The mRNA expression levels of CPT1B and CPT1C in rectal adenocarcinoma (READ) and colon adenocarcinoma (COAD) based on data from the TCGA database. A. CPT1B expression in READ. B. CPT1B expression in COAD. C. CPT1C expression in READ. D. CPT1C expression in COAD.

- Figure 2 lacks a clear description of how IHC scores were determined and the criteria used to categorize patients into CPT1A-high and CPT1A-low groups. This should be detailed in the text and figure legend.

We thank the reviewer for the valuable comment. We have provided a detailed description of the methodology used to determine the IHC scores and criteria applied to categorise patients into CPT1A-high and CPT1A-low groups in the Materials and Methods section (Lines 418-426) as well as the legend of Figure 2A.

- None of Figure 2B or 2C show how many patients were assigned to the CPT1A-low and CPT1A-high groups.

We thank the reviewer for the valuable comment. We have added the number of patients in the CPT1A-low and CPT1A-high groups to the legends of Figures 2B and 2C.

(2) Model Selection and Experimental Approaches- The authors primarily use CPT1A knockout (KO) HCT116 cells and CPT1A overexpression (OE) SW480 cells for their experiments, which poses major limitations.

We thank the reviewer for the valuable comment.

- The genetic backgrounds of the cell lines (e.g., HCT116 being microsatellite instable (MSI) and SW480 not) should be considered as they might influence treatment outcomes. This should be acknowledged as a major limitation.

We thank the reviewer for the valuable comment. Indeed, the genetic background differences among cell lines represent a significant limitation. We have addressed this issue in the discussion section (Lines 363-365).

- Regardless of their CPT1A expression levels, for the experiments with HCT116 and SW480 cells in Figure 3C-F, it would be useful to see whether HCT116 cells can be further sensitized to radiotherapy in overexpression and whether SW480 cells can be desensitized through CPT1A KO.

We thank the reviewer for the valuable comment. Due to the inherently high levels of CPT1A in the HCT116 cell line, we attempted to perform relevant experiments but were unable to achieve significant overexpression. Similarly, we faced challenges with the SW480 cell line, which has lower levels of CPT1A. We could thus not provide additional insights in this respect.

- The use of only two CRC cell lines is insufficient to draw broad conclusions. Additional CRC cell lines should be used to validate the findings and account for genetic heterogeneity. The authors should repeat key experiments with additional CRC cell lines to strengthen their conclusions.

We thank the reviewer for the valuable comment. To address this issue, we used a radiation-resistant variant of the HCT-15 cell line as a new approach to investigate whether CPT1A is associated with cellular radiation sensitivity. We believe that the data obtained from these acquired resistant cell lines are comparable to those from the ordinary cell lines mentioned in the reviewer’s comment.

(3) Pharmacological InhibitionSeveral studies have reported beneficial outcomes of using CPT1 pharmacological inhibition to limit cancer progression (e.g., PMID: 33528867, PMID: 32198139), including its application in sensitization to radiation therapy (PMID: 30175155). Since the authors argue for the opposite case in CRC, they should show this through pharmacological means such as etomoxir and whether CPT1A inhibition phenocopies their observed genetic KO effect, which would have important implications for using this inhibitor in CRC patients.

We thank the reviewer for the valuable comment. The referenced literature has indeed attracted our attention. Our research group is concurrently investigating the role of CPT1A in tumor radiotherapy and immunology, utilising CPT1A inhibitors for experimental validation. We look forward to publishing these related studies to further support the conclusions presented in our manuscript.

(4) Data Representation and Statistical Analysis- The relative mRNA expression levels across the seven cell lines (Supplementary Figure 1C) differ greatly from those reported in the DepMap (https://depmap.org/portal/). This discrepancy should be addressed.

We thank the reviewer for the valuable comment. The observed differences in mRNA levels may be attributable to variations in cell culture density. For subsequent radiation sensitivity experiments, we maintained our cell culture density at approximately 70–80% confluence.

- The statistical significance of differences in mRNA and protein levels between RT-sensitive and RT-resistant cells should be shown (Supplementary Figure 1C, D).

As suggested, we have included a statistical analysis of the differences in CPT1A mRNA levels between RT-sensitive and -resistant cells in Figure 3 and Supplementary Figure 1C. However, further analysis revealed no significant difference in CPT1A protein levels between these groups. This was attributed to the high variability in grayscale values observed between the groups.

ConclusionThe study offers significant insights into the role of CPT1A in CRC radioresistance, proposing a tumor-suppressive function. However, the scope and depth of the analysis need to be expanded to fully validate these claims. Additional CRC cell lines, pharmacological inhibition studies, and a more detailed analysis of patient samples are essential to strengthen the conclusions.

We would like to thank Reviewer #2 for the comments.

**Reviewer #3 (Public Review):**
Summary:The study aims to elucidate the role of CPT1A in developing resistance to radiotherapy in colorectal cancer (CRC). The manuscript is a collection of assays and analyses to identify the mechanism by which CPT1A leads to treatment resistance through increased expression of ROS-scavenging genes facilitated by FOXM1 and provides an argument to counter this role, leading to a reversal of treatment resistance.Strengths:The article is well written with sound scientific methodology and results. The assays performed are well within the scope of the hypothesis of the study and provide ample evidence for the role of CPT1A in the development of treatment resistance in colorectal cancer. While providing compelling evidence for their argument, the authors have also rightfully provided limitations of their work.

We would like to thank Reviewer #3 for the positive comments.

Weaknesses:The primary weakness of the study is acknowledged by the authors at the end of the Discussion section of the manuscript. The work heavily relies on bioinformatics and in vitro work with little backing of in vivo and patient data. In terms of animal studies, it is to be noted that the model they have used is nude mice with non-orthotopic, subcutaneous xenograft, which may not be the best recreation of the patient tumor.

We thank the reviewer for the insightful comment. Our research group is continuing to explore the role of CPT1A in colorectal cancer radiotherapy and immunotherapy. In a new study, we used a C57BL/6 mouse model to conduct *in-vivo* experiments. Preliminary data suggest that CPT1A confers heightened radiosensitivity to immunocompetent mice. We look forward to the forthcoming publication of this ongoing research project.

**Recommendations for the authors:**

**Reviewer #1 (Recommendations For The Authors):**
The manuscript was challenging to read and contained many typographical errors and tangents that were not logically relevant to the logic of the paper. For example, in lines 365-367 the authors talk about peroxisomes being important for redox balance and that they will target peroxisomal pathways. However, the authors do not perform any experiments targeting peroxisomal pathways. So, I found myself quite perplexed. Careful proofreading of the manuscript would improve the utility for readers.

We thank the reviewer for the insightful comments. We have made several additions throughout the manuscript to include more relevant information and experimental details, thereby improving the manuscript’s logical structure and readability. As described in the text, we used the DCFH-DA probe to measure ROS levels in cells, considering that regulation of intracellular ROS levels is a major function of peroxidases. We examined the transcriptional levels, protein expression, and enzymatic activities of peroxidases such as CAT, SOD1, SOD2, and SOD3 through qPCR, Western blotting, and specific assay kits.

**Reviewer #2 (Recommendations For The Authors):**
(1) Clarification and FlowIntroduction Clarity: The introduction introduces several topics in succession without clearly connecting them. For example, the introduction of FOXM1 on Line 102 lacks clarity in its relationship to the study. Consider discussing these elements only in the discussion section to avoid confusion.

We thank the reviewer for this insightful comment. We have moved the section on FOXM1 to the discussion to enhance readability (Lines 342-348).

Explanation for Non-experts: Both the multi-target single-hit survival model and the comet assay require one sentence to explain their principles for non-experts in the field.

As suggested, we have included brief explanations of the multi-target single-hit survival model and the comet assay in the Materials and Methods section to clarify these concepts to readers not familiar with the subject (Lines 458-460 and 462-465).

(2) Specific Text Revisions- Line 302: "We transfected the CRISPR/Cas9 lentivirus into HCT 116 ... efficiency of the 2nd site was the highest" - Clarify what is meant by "second site." If you mean the second sgRNA, please use this term.

As suggested, we have revised ‘2nd’ to ‘second’ (Lines 151 and 152).

- Lines 358-359: For the results subsection "Low CPT1A levels accelerate post-radiation ROS scavenging," include an introductory sentence, such as: "To study the mechanism of low CPT1A expression in radiotherapy resistance, we conducted differential gene expression analysis between HCT116 CPT1A KO and NC cells."

As suggested, we have added an introductory sentence in the section titled ‘Low CPT1A Levels Accelerate Post-Radiation ROS Scavenging’ (Lines 215-217).

- Line 359: "The gene expression heatmap showed high consistency among replicates for both HCT 116-NC and HCT 116-KO cells (Supplementary Figure 3A)." If these are technical replicates performed on the same batch of KO or NC cells, please state this clearly.

We have added the suggested information to improve clarity (Line 218).

- Lines 360-362: "With CPT1A knockdown, we found 363 upregulated and 1290 downregulated genes (|log2(fold change)|>1 and P<0.05)." Ensure that the p-value is correct; it seems this should be q-value < 0.05.

As suggested, we have revised ‘p’ to ‘q’ (Lines 220 and 496).

- Line 363: Introduce the term "DEGs" as Differentially Expressed Genes in the main text, not just in the Materials and Methods (line 215).

As suggested, we have introduced the term "DEGs" as Differentially Expressed Genes in the main text (Lines 221-222).

- Lines 364-365: "Showing that the main enriched pathways were in peroxisomes, cell cycle nucleotide excision repair, and fatty acid degradation (Figure 5A)." The data does not support this statement. Clarify that the listed pathways are AMONG the enriched KEGG pathways.

As suggested, we have revised the relevant part in the manuscript (Lines 222-224).

- Line 370: "...following 6 Gy irradiation and 1 h of incubation with DCFH-DA (Figure 5C)." Write out the term DCFH-DA and explain it for non-experts: "a fluorescent redox probe used to detect reactive oxygen species."

As suggested, we have added a brief explanation to clarify the term for readers not familiar with the subject (Lines 230-231).

- Line 444: "CPT1A is an essential tumor suppressor." This statement has not been validated or referenced adequately.

As suggested, we have removed the sentence to improve clarity.

- Line 447: Clarify the relevance of the He, Zhang & Xu reference.

We apologise for the error and have removed the reference.

(3) Figure Improvements- Standardize Graph Labels: Ensure that graph axis labels and numbering are consistent and legible across the manuscript. For example, Figure 1A has large labels, while Figure 1B has much smaller labels. Ensure all graphs, such as 2C and 3G, have readable labels and numbering.

We thank the reviewer for the insightful comment. We have revised the labels and numbering in Figures 1B, 2C, and 3G.

- Figure 2B and 2C: Correct the x-axis label from "mouths" to "months."

We thank the reviewer for this insightful comment. We have revised the labels in Figure 2B and 2C.

- Figure 3 Legend: Clarify what is meant by "different groups of cell lines" in the legend of Figure 3. Specify whether these are single clones, pooled clones, or mixtures of cells in the text and/or figure legend.

We thank the reviewer for this insightful comment. We have updated the legend of Figure 3 to enhance clarity.

- Figures 3H and 3I: Label the blots clearly to indicate which refer to HCT116 NC and KO and which to SW480 RFP and OE.

We thank the reviewer for this insightful comment. We have revised the labels in Figure 3H and 3I.

- Supplementary Figure 2A: Describe the terms F and W in the legend.

We thank the reviewer for this insightful comment. 'F' denotes fraction and 'W' denotes week. We have updated the legend of Figure 3 and Figure 3-figure supplement 2 to improve clarity.

- Supplementary Data: Consider moving the data described in Supplementary Figure 2 to the main figures as it is among the most convincing data in the paper.

We thank the reviewer for this insightful comment. We have decided to retain this figure at its current position, as we believe the data presented provide complementary evidence supporting the conclusion discussed earlier.